



# Airborne mapping of the sub-ice platelet layer under fast ice in McMurdo Sound, Antarctica

Christian Haas[1,2,3,4], Patricia J. Langhorne[5], Wolfgang Rack[6], Greg H. Leonard[7], Gemma M. Brett[6], Daniel Price[6], Justin F. Beckers[1,8], Alex J. Gough[5]

[1]Department of Earth and Atmospheric Science, University of Alberta, Edmonton, Canada

[2]Department of Earth and Space Science and Engineering, York University, Toronto, Canada

[3]Alfred Wegener Institute for Polar and Marine Research, Bremerhaven, Germany

4Department of Environmental Physics, University of Bremen, Bremen, Germany

[5]Department of Physics, University of Otago, Dunedin, New Zealand

[6]Gateway Antarctica, University of Canterbury, Christchurch, New Zealand

[7]School of Surveying, University of Otago, Dunedin, New Zealand

[8]Canadian Forest Service, Natural Resources Canada, Edmonton, Canada

*Correspondence to*: Christian Haas (chaas@awi.de) and/or Patricia J. Langhorne (pat.langhorne@otago.ac.nz)

**Abstract.** Basal melting of ice shelves can result in the outflow of supercooled ice shelf water, which can lead to the formation of a sub-ice platelet layer (SIPL) below adjacent sea ice. McMurdo Sound, located in the southern Ross Sea, Antarctica, is well known for the occurrence of a SIPL linked to ice shelf water outflow from under the McMurdo Ice Shelf. Airborne, single frequency, frequency-domain electromagnetic induction (AEM) surveys were performed in November of 2009, 2011, 2013, 2016, and 2017 to map the thickness and spatial distribution of the landfast sea ice and underlying, porous SIPL. We developed a simple method to retrieve the thickness of the consolidated ice and SIPL from the EM inphase and quadrature components, supported by EM forward modeling, and calibrated and validated by drill-hole measurements. Linear regression of EM inphase measurements of apparent SIPL thickness and drill-hole measurements of "true" SIPL thickness yields a scaling factor of 0.3 to 0.4, and rms error of 0.47 m. EM forward modeling suggests that this corresponds to SIPL conductivities between 900 and 1800 mS/m, with associated SIPL solid fractions between 0.09 and 0.47. The AEM surveys showed the spatial distribution and thickness of the SIPL well, with SIPL thicknesses of up to 8 m near the ice shelf front. They indicate interannual SIPL thickness variability of up to 2 m. In addition, they reveal high-resolution spatial information about the small-scale SIPL thickness variability, and indicate the presence of persistent peaks in SIPL thickness that may be linked to the geometry of the outflow from under the ice shelf.



## 1 Introduction

McMurdo Sound is an approximately 55 km wide sound in the southern Ross Sea, Antarctica, located between Ross Island and the Transantarctic Mountains in Victoria Land (Figure 1a). It is bordered by the small McMurdo Ice Shelf to the South, a portion of the much larger Ross Ice Shelf. For most of the year, McMurdo Sound is covered by landfast sea ice. The fast ice is mostly composed of first-year ice which usually breaks out during the summer months (Kim et al., 2018). However, in some years some smaller regions of fast ice mostly near the coast or ice shelf edge may persist through one or several summers to form thick multiyear landfast ice. In particular between 2003 and 2011 the southern parts of McMurdo Sound remained permanently covered by thick multiyear ice that had initially formed due to the shelter from swell and currents by the large grounded iceberg B15 further north (Robinson and Williams, 2012; Brunt et al., 2006; Kim et al, 2018).

McMurdo Sound is characterized by intensive interaction between the ice shelf, sea ice, and ocean (Gow et al., 1998; Smith et al., 2001; Leonard et al., 2011; Robinson et al., 2014; Langhorne et al., 2015). In particular, melting at the base of the Ross and McMurdo ice shelves results in the seasonally variable presence of supercooled Ice Shelf Water (ISW). A plume of supercooled ISW emerges from the McMurdo Ice Shelf and spreads north (Leonard et al., 2011; Robinson et al., 2014), leading to the widespread formation and accumulation of frazil ice to form a sub-ice platelet layer under the fast ice (Figure 1). This sub-ice platelet layer (SIPL) is a poorly consolidated, highly porous layer of millimeter- to decimeter-scale planar ice crystals (Hoppmann et al., 2020), and is an important contributor to the sea ice mass balance in McMurdo Sound and along the coast of Antarctica in general (Smith et al., 2001; Gough et al., 2012; Langhorne et al., 2015). Its presence and thickness are important indicators of the occurrence of ISW near the ocean surface (Lewis and Perkin, 1985). Subsequently, the SIPL may consolidate and become incorporated into the solid sea ice cover to form so-called incorporated platelet ice (Gow et al., 1998; Smith et al., 2001; Hoppmann et al., 2020). Due to the contributions of platelet ice, sea ice thicknesses in Antarctic near-shore environments can be larger than in the pack ice zone further offshore (Gough et al, 2012).

The SIPL in McMurdo Sound, its dependence on ocean processes, and its role for increasing sea ice freeboard and thickness has been extensively studied for many years (e.g. Gow et al., 1998; Mahoney et al., 2011; Robinson et al., 2014; Price et al., 2014; Langhorne et al., 2015; Brett et al., 2020). The spatial distribution of supercooled water and platelet ice have been observed by means of local CTD and drill-hole measurements on the fast ice (e.g. Lewis and Perkin, 1985; Barry 1988; Dempsey et al., 2010; Leonard et al., 2011; Mahoney et al., 2011; Robinson et al., 2014; Langhorne et al., 2015). These studies showed that a SIPL primarily occurs in a 20 to 30 km wide, 40 to 80 km long region extending from the northern tip of the McMurdo Ice Shelf in a northwesterly direction (Fig. 1), and locally near the coast of Ross Island. Drill-hole measurements showed that at the end of the winter the thickness of the SIPL under first-year ice can be up to 7.5 m (Price et al., 2014; Hughes et al., 2014), coinciding with more than 2.5 m of consolidated sea ice. With multiyear ice, SIPL and consolidated ice thicknesses can be much larger, depending on location and increasing with age.

Electromagnetic induction sounding (EM) measurements are sensitive to the presence of layers of different electrical conductivity in the sub surface. The presence and thickness of the porous, seawater-saturated SIPL can be retrieved because



its electrical conductivity is in between those of the resistive, consolidated sea ice above and the conductive seawater below. The technical and logistical difficulties of on-ice and drill-hole measurements often only allow discontinuous, widely spaced
sampling. Therefore observations of the kilometer-scale and interannual variability of SIPL occurrence and thickness are still rare or restricted to regions that are accessible by on-ice vehicle (Hoppmann et al., 2015; Hunkeler et al., 2015a,b; Brett et al, 2020). Notably Hunkeler et al. (2015a,b), Irvin (2018), and Brett et al. (2020) have already demonstrated the capability of ground-based, single- and multifrequency EM measurements to map the occurrence and thickness of the SIPL, using numerical inversion methods. The two latter studies have successfully reproduced the geometry of the SIPL known from earlier studies
(Barry, 1988; Dempsey et al, 2010; Langhorne et al, 2015). Using four years of ground-based EM data, Brett et al. (2020) have studied the interannual SIPL variability and found that the SIPL was thicker in 2011 and 2017 than in 2013 and 2016, in close relation to nearby polynya activity that contributes to variations in ocean circulation under the ice shelf. In spite of progress, details are lacking and the processes involved in the outflow of ISW from under the McMurdo ice shelf are still little known. In contrast to ground-based EM measurements where the instrument height over the snow or ice surface is constant, instrument
height varies significantly with AEM measurements due to unavoidable altitude variations of the survey aircraft. This makes the application of numerical inversion methods more complicated. Further the development and calibration of the empirical AEM SIPL retrieval algorithm requires that the electrical conductivity of the SIPL is known. The SIPL is an open matrix of loosely-coupled crystals in approximately random orientations, and its conductivity depends on its solid fraction $\beta$ (e.g., Gough et al., 2012; Langhorne et al, 2015), both of which are hard to measure directly. Observations and modeling over the
past four decades suggest that $\beta$ is quite low, with a mean of $\beta = 0.25 \pm 0.09$ (Langhorne et al, 2015) and range of $0.15 - 0.45$ (e.g. Hoppmann et al., 2020).

In this paper we develop a simple empirical algorithm for the joint retrieval of SIPL and consolidated ice thicknesses from single-frequency AEM measurements that is supported by an EM forward model and calibrated and validated by coincident drill-hole measurements. We show that the SIPL conductivity, and therefore its porosity or solid fraction, can be obtained as
a first step in the calibration of the method with drill-hole data. We apply this algorithm to five surveys carried out in November of 2009, 2011, 2013, 2016, and 2017 by helicopter and fixed-wing aircraft. Using these techniques we demonstrate the ability of airborne electromagnetic induction (AEM) measurements to map the small-scale distribution of the SIPL under landfast sea ice with high spatial resolution. We apply AEM to study the interannual variability of the SIPL in McMurdo Sound from which we infer some previously unknown features of the ISW plume.



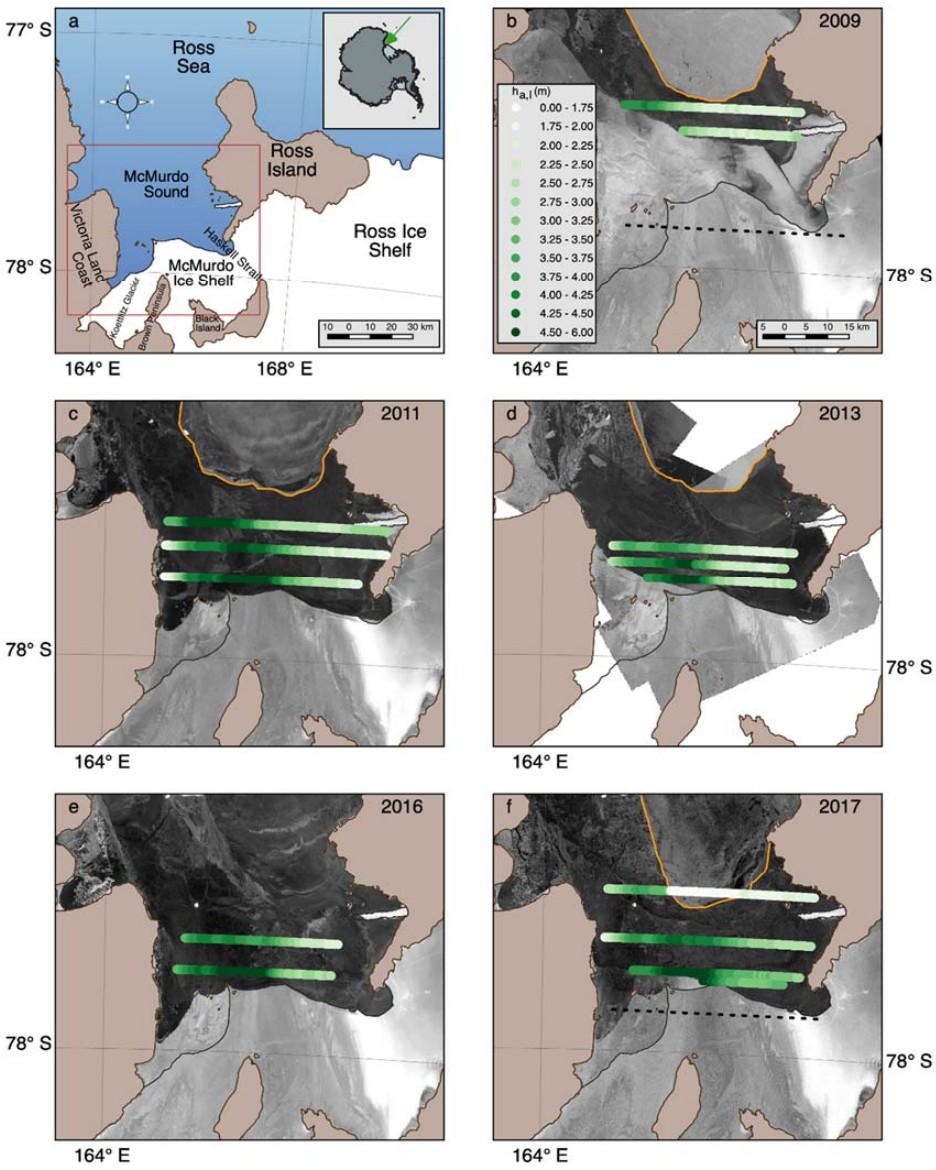

**Figure 1: Overview maps of the AEM surveys carried out in 2009, 2011, 2013, 2016, and 2017. (a) regional overview and the location of McMurdo Sound (green arrow), and boundaries of satellite images (red). (b-f) Locations of East-West profiles, overlaid on synthetic-aperture radar (SAR) satellite images to show differences in general ice conditions and ice types (Brett et al., 2020; 2009/11: Envisat; 2013: TerraSAR-X; 2016/17: Sentinel-1). Colors correspond to different apparent ice thicknesses $h_{a,I}$ (Section 2.1.2). Orange lines mark respective fast ice edges. Bright areas to the South are the McMurdo Ice Shelf. Black dashed lines in b and f show tracks of ice shelf thickness surveys used in Figure 10a and 11 (Rack et al., 2013).**



## 2 Methods and Measurements

### 2.1 AEM thickness surveys

All measurements presented here were performed with a towed EM instrument (EM Bird) suspended below a helicopter or
fixed-wing airplane, and are thus named airborne EM (AEM) surveys. The EM Bird was flown with an average speed of 80
to 120 knots at mean heights of 16 m above the ice surface (Haas et al., 2009; Haas et al., 2010). The instrument operated in
vertical dipole mode with a signal frequency of 4060 Hz and a spacing of 2.77 m between transmitting and receiving coils
(Haas et al., 2009). The sampling frequency was 10 Hz, corresponding to samples every 5 to 6 m depending on flying speed.
A Riegl LD90 laser altimeter was used to measure the Bird's height above the ice surface, with a range accuracy of +/- 0.025
m. Positioning information was obtained with a Novatel OEM2 differential GPS with a position accuracy of 3 m (Rack et al.,
2013). Details of EM ice thickness sounding are explained in the following sections.

We have carried out five surveys over the fast ice in McMurdo Sound, in November of 2009, 2011, 2013, 2016, and 2017 (Fig.
1), i.e. in the end of winter when ice thickness was near its maximum. The surveys covered several approximately 50 km long,
East-West oriented profiles across the sound, as closely as possible from shore to shore. Although the exact number of profiles
differed every year due to weather restrictions, ice conditions, or technical issues, we have attempted to cover the same profiles
every year, and to collocate them with drill-hole measurements (Section 2.2). The profiles repeated most often were located at
latitudes of 77°40' S, 77°43' S, 77°46' S, and 77°50' S, i.e. 5.5 to 7.4 km apart (Fig. 1).

EM ice thickness measurements are affected by averaging within the footprint of the instrument, which results in the
underestimation of maximum pressure ridge thicknesses (e.g. Kovacs et al., 1995). However, the fast ice in McMurdo Sound
is mostly undeformed and level. Over such level ice without an underlying SIPL the agreement of EM thickness estimates is
within +/- 0.1 m of drill-hole measurements (Pfaffling et al., 2007; Haas et al., 2009). McMurdo Sound therefore presents ideal
conditions for EM ice thickness measurements, and the levelness of the ice allows the application of low-pass filtering to
remove occasional noise that affects measurements over thick ice without losing significant information on larger scales. Here,
we have applied a running-window, 300 point median filter corresponding to a width of 1.5 to 1.8 km to all data unless
mentioned otherwise.

However, the accuracy of 0.1 m stated above relies on accurate calibration of the EM sensor, which is typically achieved by
flying over short sections of open water (Haas et al., 2009). Unfortunately open water overflights were not possible with the
helicopter surveys between 2009 and 2013, due to safety regulations. Then the calibration could only be validated over drill-
hole measurements and may be less accurate than reported above. Only in 2017 were we able to use a Basler B67 airplane,
permitting flights over the open water in the McMurdo Sound polynya which provided ideal calibration conditions (Fig. 1f).





### 2.1.1 EM response to sea ice thickness and a sub-ice platelet layer

Frequency-domain, electromagnetic induction (EM) sea ice thickness measurements rely on the active transmission of a continuous, low-frequency, "primary" EM field of one or multiple, constant frequencies, penetrating through the resistive
snow and ice into the conductive seawater underneath. As the resistivity of cold sea ice and dry snow are approximately infinite (Kovacs and Morey, 1991; Haas et al., 1997), eddy currents are only induced in the seawater underneath. These eddy currents generate a "secondary" EM field with the same frequency as the primary EM field, but with a different amplitude and phase. The EM sensor measures the amplitude and phase of the secondary field, relative to those of the primary field, in units of parts per million (ppm) of the primary field. Amplitude and phase of the complex secondary field are usually decomposed into real
and imaginary signal components, called inphase ($I$) and quadrature ($Q$), respectively:

$I$ [ppm] = Amplitude [ppm] × cos(Phase [degrees])

$Q$ [ppm] = Amplitude [ppm] × sin(Phase [degrees])

With negligible sea ice and snow conductivities, measured $I$ and $Q$ of the relative secondary field depend on the distance between the EM instrument and the ice-water interface, and on the conductivity of the seawater. With known seawater
conductivity, $I$ and $Q$ decay as an approximate negative-exponential with increasing distance ($h_0 + h_i$) between the EM instrument and the ice-water interface, where $h_0$ is instrument height above the ice and $h_i$ is ice thickness (see Section 2.1.3 below; Haas, 2006; Pfaffling et al., 2007; Haas et al., 2009):

$$I \approx c_0 \times \exp(-c_1 \times (h_0+h_i)) \tag{1a}$$

$$Q \approx c_2 \times \exp(-c_3 \times (h_0+h_i)) \tag{1b}$$

with constants $c_{0..3}$. Then, height above the ice-water interface can be obtained independently from both, $I$ and $Q$, from equations of the form:

$$(h_0+h_i) \approx -1/c_1 \times \ln(I/c_0) \tag{2a}$$

or

$$(h_0+h_i) \approx -1/c_3 \times \ln(Q/c_2) \tag{2b}$$

For ground-based measurements with an EM instrument located on the snow or ice surface (i.e. $h_0 = 0$ m), the distance to the ice-water interface corresponds to the total (snow-plus-ice) thickness $h_i$ (Kovacs and Morey, 1991; Haas et al., 1997). With airborne measurements, the variable height of the EM instrument above the snow or ice surface $h_0$ is measured with a laser altimeter. Then, total (ice-plus-snow) thickness is computed from the difference between the electromagnetically derived height above the ice-water interface and the laser determined height above the snow or ice surface:

$$h_{i,I} \approx -1/c_1 \times \ln(I/c_0) - h_0 \tag{3a}$$

or

$$h_{i,Q} \approx -1/c_3 \times \ln(Q/c_2) - h_0 \tag{3b}$$

(Pfaffling et al., 2007; Haas et al., 2009). Over typical, saline sea water $I$ (Eq. 1a) is two to three times larger than $Q$ (Eq. 1b) and has much better signal-to-noise characteristics (Haas, 2006). It is therefore the preferred channel for ice thickness retrievals



160 in the Arctic and Antarctic (Haas et al., 2009). Because snow and ice are indistinguishable for EM measurements due to their low conductivity, no attempt is made here to distinguish between them, and the terms ice thickness or consolidated ice thickness are used throughout to describe total, i.e. snow plus ice thickness.

### 2.1.2 Apparent thickness

In the case of the presence of a SIPL, ice thickness retrievals become significantly more difficult, and will lead to large errors
165 if the effect of the SIPL is not taken into account. Induction in the conductive SIPL results in an additional secondary field which mutually interacts with the secondary field induced in the water underneath. Thus the EM signal becomes a function of both, consolidated ice thickness, and the thickness and conductivity of the SIPL. The conductivity of the porous SIPL is higher than that of consolidated ice (~0 mS/m) but most likely lower than that of the seawater underneath (approximately 2700 mS/m in McMurdo Sound; e.g. Mahoney et al., 2011; Robinson et al., 2014). Therefore, over consolidated ice underlain by a SIPL,
170 the measured secondary field will be smaller than if there was no SIPL, and larger than if the SIPL was consolidated throughout and highly resistive. Using the simple, negative-exponential relation between $I$ or $Q$ and ice thickness described above (Eqs. 1, 3), smaller $I$ and $Q$ due to the presence of a SIPL will result in consolidated ice thickness estimates $h_a$ that are larger than the true consolidated ice thickness $h_i$. However, the derived consolidated ice thickness $h_a$ will be less than the total ice plus SIPL thickness $h_i + h_{sipl}$ because the thickness retrieval assumes negligible ice conductivity, which is an invalid assumption
175 for the SIPL. Therefore the measured $I$ and $Q$ would be larger than they are for negligible SIPL conductivity. Here we introduce the term "apparent thickness", $h_a$, to describe the ice thickness obtained from either $I$ (or $Q$) following the standard procedures and simple negative exponential relationship in Eq. 3 (Haas et al., 2009; Rack et al., 2013). This is the thickness that one obtains if the presence of a SIPL was not considered. The apparent thickness $h_a$ agrees with the true thickness $h_i$ if the ice has negligible conductivity. Otherwise, in the presence of a conductive SIPL, the apparent thickness $h_a$ will be more than the
180 consolidated ice thickness, but less than the total, consolidated ice plus SIPL thickness $h_i + h_{sipl}$.

As will be shown in Section 2.1.3, $I$ and $Q$ respond differently to the presence of a SIPL, and Q is in fact little affected and can therefore still be used to retrieve $hi$. The presence of this layer can therefore be detected by deviations between the apparent thicknesses derived from I and Q. The different responses of $I$ and $Q$ can also be used to determine the thickness of the SIPL, and thus to convert apparent thickness into consolidated ice and SIPL thicknesses (Section 2.1.4). In general, the thickness and
185 conductivity of consolidated ice and the SIPL can be derived by means of full, least-square layered-earth inversion of airborne I, Q, and laser altimeter data, and by potentially using more signal frequencies (e.g. Rossiter and Holladay, 1994; Pfaffling and Reid, 2009; Hunkeler et al, 2015a,b). However, numerical inversion is computationally demanding and requires well calibrated data with good signal-to-noise characteristics. In addition, these algorithms require certain a-priori knowledge about the stratigraphy of the ice, i.e. layers present and their conductivities. The development and application of such algorithms is
190 beyond the scope of this paper. Instead, here we apply a simple empirical algorithm for the joint retrieval of SIPL and





consolidated ice thicknesses from single-frequency AEM measurements. The following section will outline the theoretical basis for this approach, including results from an EM forward model and a discussion of assumptions that need to be made.

### 2.1.3 Modeling EM responses over fast ice with a SIPL

To demonstrate the sensitivity of EM measurements to the presence of a SIPL, and to evaluate the potential of determining its
thickness, we performed extensive one-dimensional forward modeling of the EM response to different SIPL thicknesses and conductivities. The $I$ and $Q$ components of the complex relative secondary field measured with horizontal coplanar coils over $n$ horizontally stratified layers overlying a homogeneous half-space can be calculated as (e.g. Mundry, 1984; Guptasarma and Singh 1997):

$$(I + jQ) = r^2 \int_0^\infty \lambda^2 R_0 e^{-2\lambda h_0} J_0(\lambda r) d\lambda \tag{4}$$

This is a so-called Hankel transform utilizing a Bessel function of the first kind of order zero ($J_0$), with $r$ being the coil spacing, $h_0$ the receiver and transmitter height above the ice and $\lambda$ the vertical integration constant. $R_0$ is called transverse electric reflection coefficient and is a recursive function of system angular frequency $\omega$ and the thickness and electromagnetic properties of individual layers (electrical conductivity $\sigma$, and magnetic permeability $\mu_0$):

$$R_{n-1} = K_{n-1}$$
$$R_{k-2} = (K_{k-2} + R_{k-1} \, u_{k-1})/(1 + K_{k-2} \, R_{k-1} \, u_{k-1})$$

with

$$u_k = \exp(-2h_k \, v_k)$$
$$v_k = (\lambda^2 + j\omega\mu_0\sigma_k)^{1/2}$$
$$K_{k-1} = (v_{k-1} - v_k)/(v_{k-1} + v_k)$$

In these equations $n$ is the number of layers (4 in this paper: non-conductive air, sea ice and snow, SIPL, and seawater), $k = 1$ (air),.., 4 (seawater), and $j$ = sqrt(-1). Figure 2 shows the general design of this 4-layer case, and also the layer properties used for the computations which are based on the typical conditions during our surveys in McMurdo Sound.

As the EM signal is ambiguous for variable layer thicknesses and conductivities, we only calculate signal changes due to variable SIPL (layer 2) thicknesses $h_{sipl}$ and conductivities $\sigma_{sipl}$, keeping all other parameters constant and representative of our
measurements: we chose instrument height $h_0$ = 16 m, sea ice (layer 1) thickness $h_i$ = 2 m, and seawater (layer 3: infinitely deep, homogeneous half-space) conductivity $\sigma_w$ = 2700 mS/m. SIPL conductivity $\sigma_{sipl}$ was varied between 0 and 2700 mS/m in steps of 300 mS/m to study a range of properties between the two extreme cases of negligible and maximum, seawater conductivity. SIPL thickness $h_{sipl}$ was varied from 0 to 20 m to also include the most extreme potential cases, such as to investigate the EM signal behaviour over potentially thick platelet layers under multiyear ice or an ice shelf.



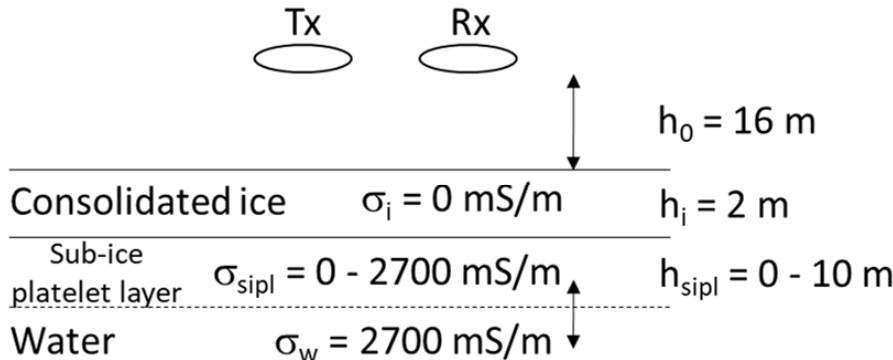


**Figure 2: Schematic of the four-layer forward model to compute EM responses over sea ice underlain by a SIPL with variable thickness and conductivity. Tx and Rx illustrate transmitting and receiving coils, respectively. Instrument height $h_0$ = 16 m, snow + ice thickness $h_i$ = 2 m, and water conductivity of 2700 mS/m are based on typical conditions during our surveys in McMurdo Sound.**

Figure 3 shows the dependence of $I$ and $Q$ model curves over 2 m thick consolidated ice on variable SIPL thickness and conductivity obtained using the model of Eq. 4. As expected it can be seen that $I$ and $Q$ do not change with increasing SIPL thickness if the SIPL conductivity is 2700 mS/m, i.e. if the SIPL is indistinguishable from sea water. $I$ decreases exponentially with increasing SIPL thickness, the effect becoming more pronounced as SIPL conductivity decreases. When SIPL conductivity is 0 mS/m, i.e. when the SIPL is indistinguishable from consolidated ice, the resulting curve is identical to

measurements over consolidated ice only, i.e. generally following the form of Eq. 1.

In contrast, and not quite intuitively, initially $Q$ changes little with increasing SIPL conductivity and thickness. Indeed, $Q$ even increases slightly with increasing SIPL thickness if the SIPL conductivity is high (e.g. larger than 600 mS/m). Only for very low SIPL conductivity (e.g. below 600 mS/m) does $Q$ decrease strongly, and for a conductivity of 0 mS/m the curve is identical to the consolidated ice case, as for $I$. Note that $I$ is generally much larger than $Q$, and that $I$ is more strongly dependent on SIPL

thickness. Therefore the sensitivity of $I$ to the presence, thickness, and conductivity of a SIPL is much larger than that of $Q$.



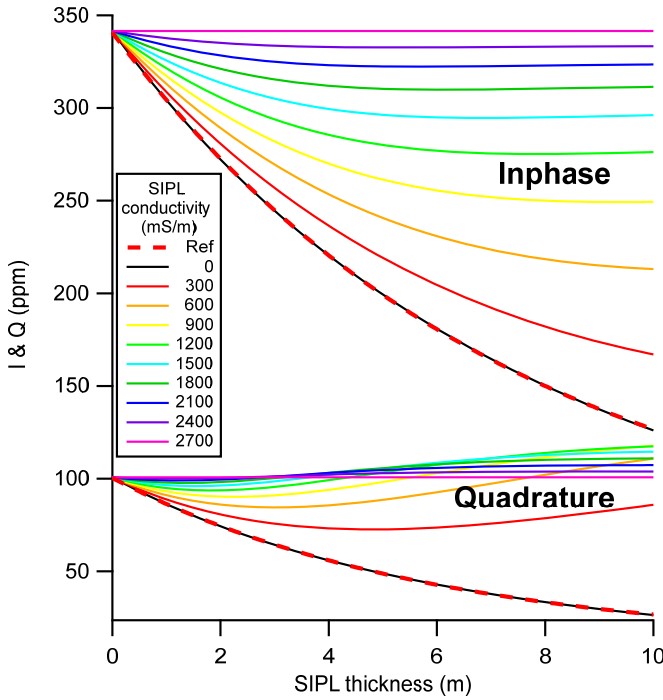

**Figure 3: Inphase *I* and Quadrature *Q* responses to a 0 to 10 m thick SIPL under 2 m thick consolidated ice for SIPL conductivities of 0 to 2700 mS/m computed with three-layer EM forward model (see Fig. 2). Ref shows negative exponential curves for consolidated ice with zero conductivity used for computation of *I* and *Q* apparent thicknesses using Eq. 2b.**


Figure 4 shows the apparent thicknesses, $h_{a,I}$ and $h_{a,Q}$, that result from applying Eqs. 3a & b to the *I* and *Q* curves in Figure 3. Eqs. 3a & b correspond to a SIPL conductivity of 0 mS/m, that would be used if the presence of an SIPL would be unknown or ignored. For example, and based on the same reasoning as above, Figure 4 shows that the apparent thicknesses agree with the total thickness $h_i + h_{sipl}$ if the SIPL conductivity was zero, i.e. indistinguishable from solid ice. If the conductivity of the

SIPL was indistinguishable from that of seawater (i.e. 2700 mS/m), the obtained apparent thicknesses are 2 m, i.e. the thickness of the consolidated ice only. For the inphase component, Figure 4a shows that apparent thicknesses for intermediate SIPL conductivities fall in between, with increasing apparent thicknesses with decreasing SIPL conductivities.

In contrast, apparent thicknesses derived from *Q* (Fig. 4b) are similar to the consolidated ice thickness (2 m in this case) for most SIPL conductivities. Only for SIPL conductivities below 600 mS/m are there relatively stronger deviations, and for a

SIPL conductivity of 0 mS/m the Quadrature derived apparent conductivity equals the total thickness $h_i + h_{sipl}$. In summary these results show that the inphase signal *I* responds much more strongly to the presence of a SIPL than the quadrature *Q*.


Note that most inphase curves level out for large SIPL thicknesses, the effect being exacerbated for higher SIPL conductivities (Fig. 3). This is due to the limited penetration depth of EM fields in highly conductive media. Accordingly the corresponding, derived apparent conductivities level out with increasing SIPL thickness as well and are insensitive to further increases of

SIPL thickness (Fig. 4a). In practice this means that the EM inphase signals are only sensitive to SIPL thickness changes up to a certain SIPL thickness, and that the sensitivity decreases with increasing SIPL thickness and conductivity. In contrast, while $Q$ is relatively insensitive to the presence and thickness of a SIPL for SIPL conductivities above 600 mS/m, responses are non-monotonic for low SIPL conductivities and possess local minima at varying SIPL thicknesses. As a result, apparent thicknesses derived from $Q$ possess local maxima at variable SIPL thicknesses.

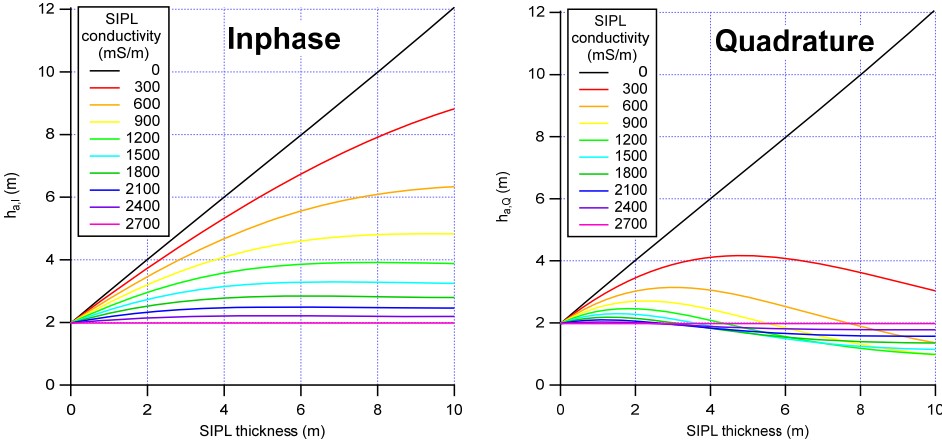

**Figure 4: Apparent thicknesses resulting from applying simple negative-exponential equations like in Eq. 3a&b to the $I$ and $Q$ curves in Figure 3.**

### 2.1.4 SIPL and consolidated ice thickness retrieval from measurements of $I$ and $Q$

The contrasting behavior of $I$ and $Q$ to variable SIPL thickness and conductivity (Fig. 3), and the resulting contrasting behavior of the derived apparent thicknesses (Fig. 4) can be used to retrieve SIPL and consolidated ice thicknesses. The figures show that, if we derive apparent thicknesses from both the $I$ and $Q$ measurements independently, the results will agree if there is just consolidated ice under the EM instrument, and they will disagree if there is a SIPL under the consolidated ice. In general, the disagreement between $h_{a,Q}$ and $h_{a,I}$ will be larger the thicker the SIPL is. In other words, the presence and thickness of a SIPL

can be retrieved from $I$ and $Q$ measurements, within limits.

Using the behavior described above, we derive the thickness of the consolidated ice $h_i$ directly from the apparent conductivity of the $Q$ measurement $h_{a,Q}$, as it is mostly insensitive to the presence of a SIPL (Fig. 4b):

$$h_i = h_{a,Q} \tag{5a}$$





Then, according to Figure 4a the apparent thickness derived from the inphase measurements $h_{a,I}$ corresponds approximately to
the sum of consolidated ice thickness and a fraction $\alpha$ of the true SIPL thickness:

$$h_{a,I} = h_i + \alpha\, h_{sipl} \approx h_{a,Q} + \alpha\, h_{sipl} \qquad\qquad (5b)$$

Therefore we can derive $h_{sipl}$ from:

$$h_{sipl} = (h_{a,I} - h_i) / \alpha \approx (h_{a,I} - h_{a,Q}) / \alpha \qquad\qquad (5c)$$

The SIPL scaling factor α primarily depends on the SIPL conductivity and governs how much the true SIPL thickness is
underestimated (Fig. 4a). The expected range of α values in Eq. 5c, and the uncertainty resulting from Eq. 5a are shown in
Figure 5.

Figure 5a shows the ratio of apparent thickness $h_a$ over "true" consolidated ice thickness $h_i$ which should be 1 according to Eq.
5a. However, it can be seen that the ratio strongly depends on $h_i$ and SIPL conductivity. In general the ratio is larger than 1 for
a thin SIPL, and smaller than 1 for a thick SIPL. The deviations from 1 decrease with increasing $h_i$, and with increasing SIPL
conductivity. For example, for $h_i = 2$ m and a SIPL conductivity of 1200 mS/m the ratio first increases to 1.27, and then
decreases to a minimum of 0.7 before slowly increasing again (Fig. 5a). This means that with a true consolidated ice thickness
of 2 m, typical for end-of-winter first-year fast ice in McMurdo Sound, our method (Eq. 5a) overestimates or underestimates
the true consolidated ice thickness by up to 30 %. However, the actual uncertainty depends on SIPL thickness, and decreases
with increasing $h_i$.
Fig. 5b shows that α (Eq. 5c) decreases monotonically with increasing SIPL thickness and conductivity. For example, for a
SIPL conductivity of 1200 mS/m it decreases from a value of 0.55 with no SIPL to values below 0.1 for a very thick SIPL
with $h_{sipl}$>>15 m. At a SIPL conductivity of 1200 mS/m it ranges between α = 0.4 and 0.3 for SIPL thicknesses between 3.7
and 6.2 m. There is little dependence on consolidated ice thickness $h_i$. These results imply that the uncertainties due to unknown
SIPL thickness (the parameter that should actually be derived from this procedure) and SIPL conductivity can be quite large.
This is because of the increasingly limited sensitivity of the AEM measurements to increasing SIPL thicknesses discussed
above with regard to Figure 4a and penetration depth.





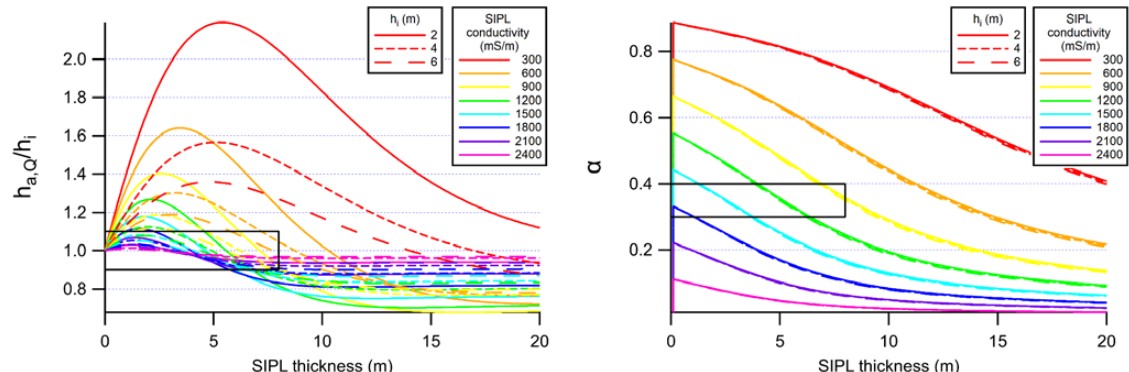

**Figure 5: Ratio of $h_{a,Q}/h_i$ (left; see Eq. 5a) and $\alpha = (h_{a,I} - h_i)/h_{sipl}$ (right; see Eq. 5c) versus SIPL thickness, at different consolidated ice thicknesses $h_i$ between 2 and 6 m, and different SIPL conductivities between 300 and 2400 mS/m. Curves follow from curves in Figures 3 and 4. Black boxes show the range of $h_{a,Q}/h_i$ and $\alpha$ values resulting from the calibration (Section 3.2).**

## 2.2 Drill-hole validation measurements

At 55 sites over all five years of observation, drill-hole measurements were performed under the flight tracks of the EM Bird to measure the thickness of snow, sea ice, and the SIPL, and the freeboard of the ice. The protocol at each drill site has been described in Price et al. (2014) and Hughes et al. (2014). At each site, five measurements were made at the centre and corners of a 30 m wide "cross". Sea ice thickness and the depth of the bottom of the sub-ice platelet layer were measured with a classical T-bar at the end of a tape measure lowered through the ice and pulled up until resistance was felt (Haas and Druckenmiller, 2009; Gough et al., 2012). This is an established method in the absence of a sub-ice platelet layer, with ice thickness accuracies of 2 to 5 cm. However, the bottom of the unconsolidated sub-ice platelet layer is often fragile and may be gradual, such that pull resistance may only increase gradually and may be difficult to feel (Gough et al., 2012). This is further complicated by the frequent presence of ice platelets inside the drill hole causing additional resistance and impeding detection of the water level within the hole. Ice crystals may be jammed between the T-anchor and the bottom of the consolidated ice, hampering the accurate determination of sea ice thickness. Following Price et al. (2014), we assume of typical relative errors (one standard deviation) for the drill-hole sea ice and sub-ice platelet layer thicknesses to be ±2% and ±5 to 30%, respectively. Snow thickness was measured on the cross lines at 0.5 m intervals using a ruler (2009, 2011, and 2013) or a Magnaprobe (2016 and 2017; Sturm and Holmgren, 2018). Throughout this paper we have added snow and ice thickness to comprise total, consolidated ice thickness $h_i$.



## 3 Results

### 3.1 Apparent ice thicknesses in McMurdo Sound

The SAR images in Figures 1 (b) – (f) show that the fast ice in McMurdo Sound can be quite variable, both with regard to the location of the ice edge and to the types of first-year ice that are present (Brett et al., 2020). Due to break-up events during the winter there can be refrozen leads with younger and thinner ice, or larger areas of thinner ice, as can for example be seen in 2013 in the northeast of the panel. These variable ice conditions result in variable thickness profiles that are indistinguishable from small undulations due to instrument drift. The SAR images also show the presence of multiyear landfast ice in some years, in particular in 2009. The multiyear ice is much thicker than the first-year ice, and we have few drill-hole measurements there. Therefore, results over multiyear ice are not included here.

Figures 1 (b) – (f) also show the apparent thickness $h_{a,I}$ determined from the inphase component along the profiles. In general it can be seen that $h_{a,I}$ ranges between 2.0 and 2.5 m, in the eastern side of the sound, in good agreement with other studies (Price et al., 2014, Brett et al., 2020), and with our drill-hole measurements (see below). On the western side of the sound much thicker ice, up to 6 m in apparent thickness can be seen. The regional distribution and thickness of this thick ice coincides with our general knowledge of the distribution of the ISW plume and the SIPL in the region (Dempsey et al., 2010; Langhorne et al., 2015). In particular, the data show that apparent ice thicknesses are much larger near the ice shelf than farther north, in agreement with the fact that the ISW plume emerges from the ice shelf and then spreads north. However, the obtained apparent thicknesses are much smaller than what is known from drill-hole measurements, when SIPL thickness is taken into account. These results confirm the results of our modeling study and demonstrate that the inphase measurements are sensitive to the presence and thickness of an SIPL.

In general, apparent thicknesses $h_{a,Q}$ derived from the quadrature measurements show much less variability. We will present them below where we show the derived consolidated ice thicknesses (Section 3.2, Fig. 6).

### 3.2 Calibration of consolidated ice and SIPL thickness

The behavior of $h_{a,Q}$ and $h_{a,I}$ can be seen much better when vertical cross sections of individual profiles are plotted. This is shown in Figure 6 for one profile near the ice shelf edge (Fig. 6a), and one farther north (Fig. 6b). The figure also shows drill-hole data for comparison. Note that here and in Figure 9 we plotted thickness downwards to illustrate more intuitively the bottom of the consolidated ice and SIPL. It can be seen that $h_{a,Q}$ and $h_{a,I}$ agree with each other quite well in the east (right) and show an ice thickness of approximately 2.0 to 2.5 m, in agreement with the consolidated ice thickness in that region. However, farther west (left), in the region of the ISW plume and thicker SIPL, the curves deviate from each other. While $h_{a,Q}$ changes relatively little, $h_{a,I}$ increases strongly. The curves join again in the farthest west, where the ISW plume is known to vanish (Robinson et al, 2014). While both curves follow the expected behaviour resulting from the model results well (Section 2.1.3), and while $h_{a,Q}$ is in reasonable agreement with the drill-hole measurements of consolidated ice thickness $h_i$, $h_{a,I}$ strongly underestimates total ice thickness $h_i+h_{sipl}$. Therefore, according to Eq. 5b, the drill-hole measurements of SIPL thickness can





be multiplied by a factor of $\alpha$ = 0.4 for best agreement with the inphase AEM measurements. Note that this behaviour and value for $\alpha$ are also in good agreement with the model results and with the range of $\alpha$ values predicted by Figure 5b.

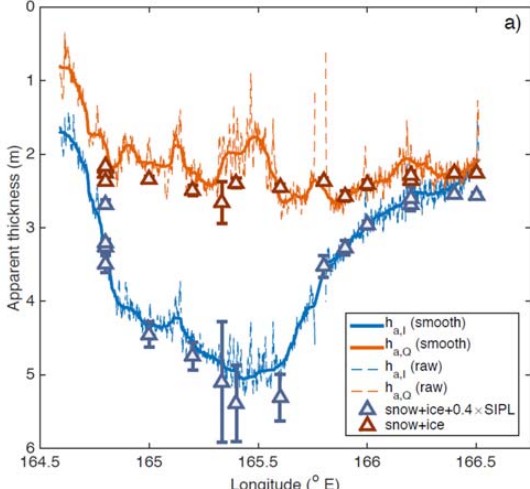


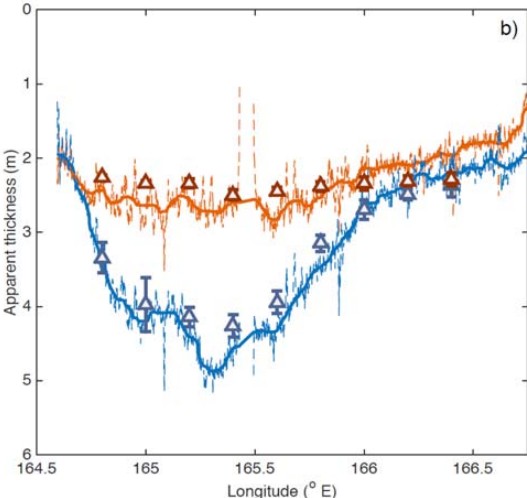

**Figure 6: Apparent AEM thicknesses $h_{a,Q}$ (orange lines) and $h_{a,I}$ (blue lines) along E-W profiles at (a) 77°50'S and (b) 77°46' S in Nov 2011, approximately 3 km and 11 km from front of McMurdo Ice Shelf, respectively. Dotted lines are raw data, while solid lines are filtered with a 300 point median filter. Triangles show mean and standard deviation of drill-hole measurements at calibration points: Consolidated ice (snow plus ice; orange), and consolidated ice + ($\alpha$ = 0.4) SIPL thickness (blue; Eq. 5c).**






The good agreement between $h_{a,Q}$ and $h_i$ from the drill-hole data strongly supports our approach of using $h_{a,Q}$ as the best estimate for $h_i$ (Eq. 5a). This approach will be evaluated below (Fig. 7). In order to determine the best values for $\alpha$, we have fit the drill-hole measured ice and SIPL thicknesses against ($h_{a,I}$- $h_{a,Q}$) measured by the EM Bird at the same sites. These values for $\alpha$ are summarized in Table 1. For first year, land-fast sea ice in 2009, 2011 and 2017 there are $N$ = 46 coincident

measurements and they yield a best fit value of $\alpha = 0.40\pm0.07$. A SIPL factor of $\alpha = 0.4$ has been used for those years henceforth. Fewer drill hole measurements were available in 2013 and 2016 ($N$ = 9 in total), resulting in a best fit of $\alpha = 0.30\pm0.15$. We therefore use an SIPL scaling factor of $\alpha = 0.3$ for 2013 and 2016 from here on.

With these $\alpha$ values we can then convert all inphase and quadrature measurements into total consolidated ice plus SIPL thickness. Figure 7 shows a scatter plot of thicknesses thus derived versus total drill-hole thicknesses. It demonstrates that EM

derived and drill-hole thicknesses agree very well, with a best fit line of slope $1.00 \pm 0.05$ and intercept $0.0 \pm 0.2$ (95% confidence intervals), and root mean square error of 0.47 m. Based on this and the discussion of Figure 5 above we also estimate that the systematic error associated with the uncertainty in the simplified processing of the $h_{a,Q}$ and $h_{a,I}$ data and the choice of $\alpha$ yields a data reduction model uncertainty of $\pm$ 0.5 m.


**Table 1:** Summary of drill-hole calibration results showing number of drill-hole measurements N, derived scaling factor $\alpha$ (Eq. 5c), and SIPL conductivity $\sigma_{SIPL}$ and solid fraction $\beta$. Data from several years with similar behavior were pooled to increase the number of data points for more reliable fits.

| Year (Nov) | $N$ | $\alpha$ (95% conf. int.) | $\sigma_{SIPL}$ (mS/m) | $\beta$ |
|---|---|---|---|---|
| 2009, 2011 & 2017 | 46 | 0.40±0.07 | 900 – 1500 | 0.16 – 0.47 |
| 2013 & 2016 | 9 | 0.30±0.15 | 1000 – 1800 | 0.09 – 0.43 |

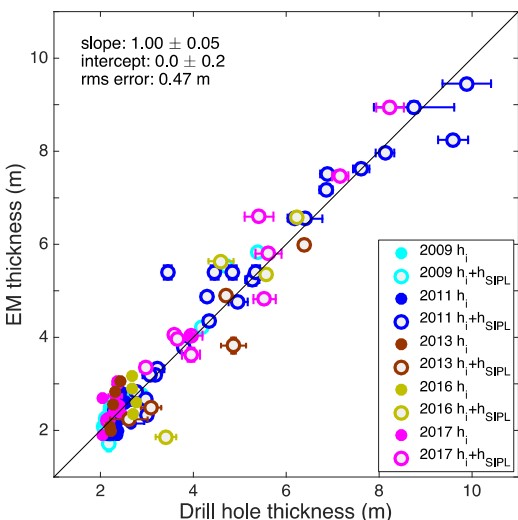

**Figure 7: Scatter plot of EM derived versus drill-hole consolidated ice thickness ($h_i$; filled symbols) and total thickness ($h_i + h_{sipl}$; open symbols), with symbol color denoting year of measurement. Total thickness ($h_i + h_{silp}$) was calculated with $\alpha = 0.4$ (best fit value 0.40±0.07, $N$=46) for 2009, 2011, 2017 and $\alpha = 0.3$ for 2013 and 2016 (best fit value 0.30±0.15, $N$=9), see Table 1. Error bars show ice thickness variability at each calibration location (5 drill holes) or within the approximate EM footprint (the latter are too small to be visible at the scale of the graph). The black line is 1:1. $N$ = 9 for 2009, $N$ = 26 for 2011, $N$ = 5 for 2013, $N$ = 4 for 2016, $N$ = 11 for 2017, thus total $N$ = 55.**

### 3.3 SIPL conductivity and solid fraction

The SIPL scaling factors, $\alpha$, are highly sensitive to SIPL conductivity and thickness (Fig. 5b). However, with known $\alpha$ and SIPL thicknesses from the drill-hole calibrations in Section 3.2 (Table 1), we can narrow down the range of possible SIPL conductivities, in particular as the range of SIPL thicknesses only extends between 0 and 8 m. The corresponding region of $\alpha$-values and SIPL thicknesses has been marked in Figure 5b. It can be seen that most curves within this region have conductivities between 900 and 1800 mS/m. The range of conductivities resulting from different $\alpha$s in the different years are listed in Table 1.

In order to relate the conductivities to a solid fraction within the SIPL we need a model of electrical conductivity, Archie's law being the best known (Archie, 1942). Figure 8 shows the horizontal conductivity for Archie's Law with cementation factor $m$=1.75 (Haas et al., 1997), and $m$=3 (Hunkeler et al., 2015b), as the solid fraction is increased from 0 to 1. For sea ice specifically, Jones et al. (2012a) have used a simple conductivity model (Jones et al., 2012b) to derive parameters of a sea ice "unit cell". They found that the parameters that fit the observed in situ DC horizontal and vertical resistivities depend not only on sea ice temperature but also on structure. In particular, for Antarctic incorporated platelet ice at -5°C the shape of the





inclusions has relative brine pocket dimensions in the horizontal $a \approx 1$, $b \approx 17$, and vertical $c \approx 0.6$ and $d \approx 6$ (see Jones et al. (2012b) for details). In addition, Jones et al (2012b) have shown for Arctic first-year sea ice that there is a dramatic change in these parameters with temperature, with $a$, $c$ and $d$ becoming larger, while $b$ drops. This behavior would also be expected in incorporated platelet ice. We shall therefore assume that the shape of the inclusions in the SIPL is similar to that of incorporated platelet ice (as observed by Jones et al., 2012a), but that brine inclusion/void dimensions are very much larger because they

are very close to the freezing point. Consequently, we calculate the relationship between solid fraction and conductivity from Jones et al. (2012b), by varying $a$ and $c$, while keeping $b$ and $d$ constant and hence changing the solid/liquid content of the SIPL (see Fig. 8).

From these curves, and the conductivities derived from the comparison of EM and drill-hole SIPL thicknesses (Fig. 5b, Table 1), we can estimate the corresponding solid fraction of the SIPL, with data in Figure 8 grouped into two sets for the range of

conductivities for $\alpha = 0.4$ in blue (2009, 2011, 2017), and $\alpha = 0.3$ in red (2013, 2016). Thus the airborne measurements imply that the range of solid fractions in the SIPL lies between 0.1 and 0.5, but values are lower in 2013 and 2016, than in 2009, 2011 and 2017 (see Table 1 and Fig. 8). We shall discuss this interannual variability in Section 3.4.1.

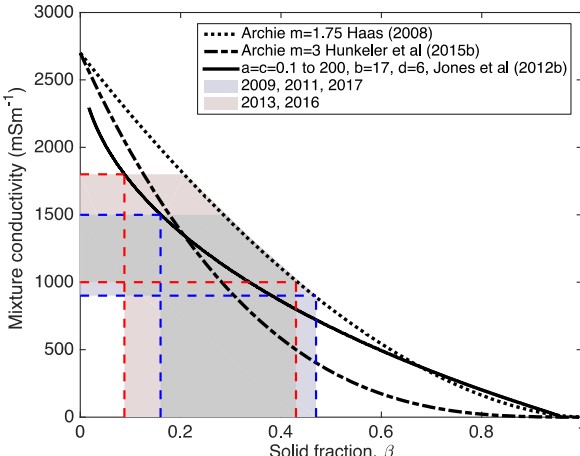

**Figure 8: Conductivity versus solid fraction for porous media according to theories by Archie (1942) with different cementation factors m = 1.75 and 3, and Jones et al. (2012b; black curves). Colored areas show range of SIPL conductivities derived from comparison of drill-hole and EM SIPL thicknesses (Fig. 5b, Table 1), and resulting solid fractions according to the different theories.**



### 3.4 Spatial and interannual variability of the SIPL

**3.4.1 Interannual variability and latitudinal differences**

Figure 9 shows the distribution of consolidated ice thickness $h_i$, and total ice thickness $h_i+h_{sipl}$ along two E-W transects in 2011, 2013, 2016, and 2017, derived from the AEM surveys and drill-hole data. The transects are 3 to 5 km (Fig. 9a) and 11 km north of the ice shelf front Fig. 9b). In 2013 and 2017 there was some multiyear ice in south-western McMurdo Sound (see Figures 1d & f), and at these locations both consolidated ice and the SIPL show abrupt increases in thickness.

The figure shows a generally thicker SIPL along the southern transect, in agreement with the notion of a thick SIPL that emerges from under the ice shelf and thins towards the north, with increasing distance from the ice shelf. Over the first-year ice, on both transects $h_i$ varies by less than 0.75 m from year to year, while variations of up to 2 m are seen in SIPL thickness $h_{sipl}$. While there is interannual variability in the thickness of the SIPL, the shape of the thickness distribution is remarkably consistent from year to year.


### 3.5 Evidence of persistent, recurring, SIPL pattern

Close inspection of the thickness data in all years and at all latitudes reveals the presence of persistent, recurring local maxima or clear shoulders in the thickness profiles. Typical examples that were identified are illustrated by (A and B) in the repeated

profiles along transect 77°46' S in 2011, 2013, 2016 and 2017 in Figure 10a. Figure 10b shows that these maxima are also present in a series of profiles at increasing latitude or decreasing distance from the front of the McMurdo Ice Shelf. While Figure 10a also shows the typical interannual variability of up to 2 m in SIPL thickness already seen in Figure 9, Figure 10b nicely demonstrates the decreasing SIPL thickness with increasing distance from the ice shelf already indicated by the differences between Figures 9a and b.


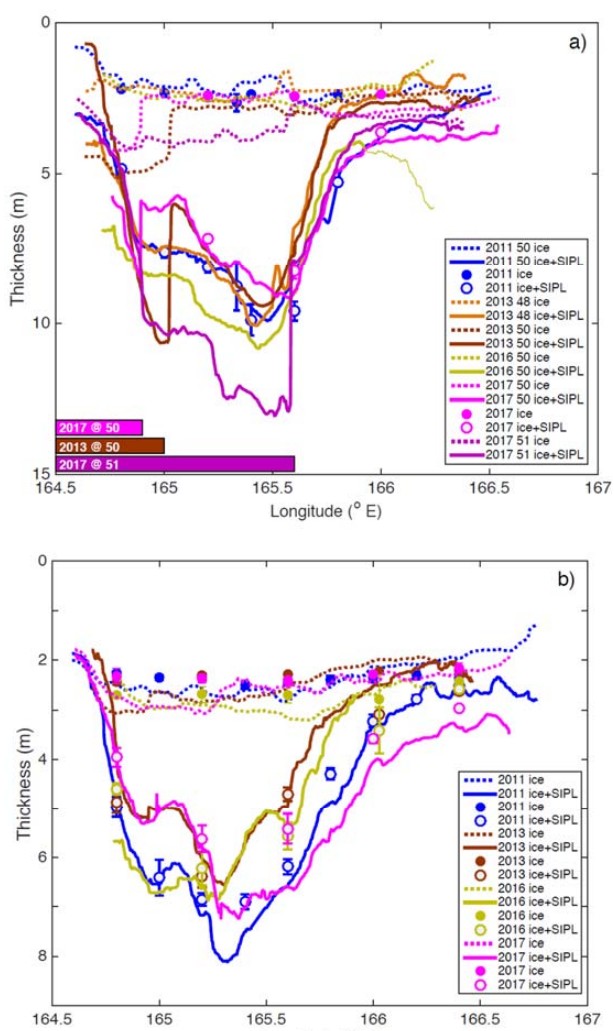

**Figure 9: Interannual variability of AEM derived and drill-hole consolidated ice $h_i$ (stippled lines and filled symbols) and total thickness $h_i+h_{sipl}$ (solid lines and open symbols) along E-W transects at similar latitude (median filtered). (a) Transects at approximately 77°48'S to 77°51'S in Nov 2011, 2013, 2016, and 2017, approximately 3 to 5 km from McMurdo Ice Shelf front. Horizontal bars indicate very thick MY ice present in the West along part of the profiles in 2013 and 2017. (b) Transect at approximately 77°46'S in Nov 2011, 2013, 2016 and 2017, approximately 11 km from McMurdo Ice Shelf front. Note different y-axis scales, i.e. thicker SIPL farther south.**



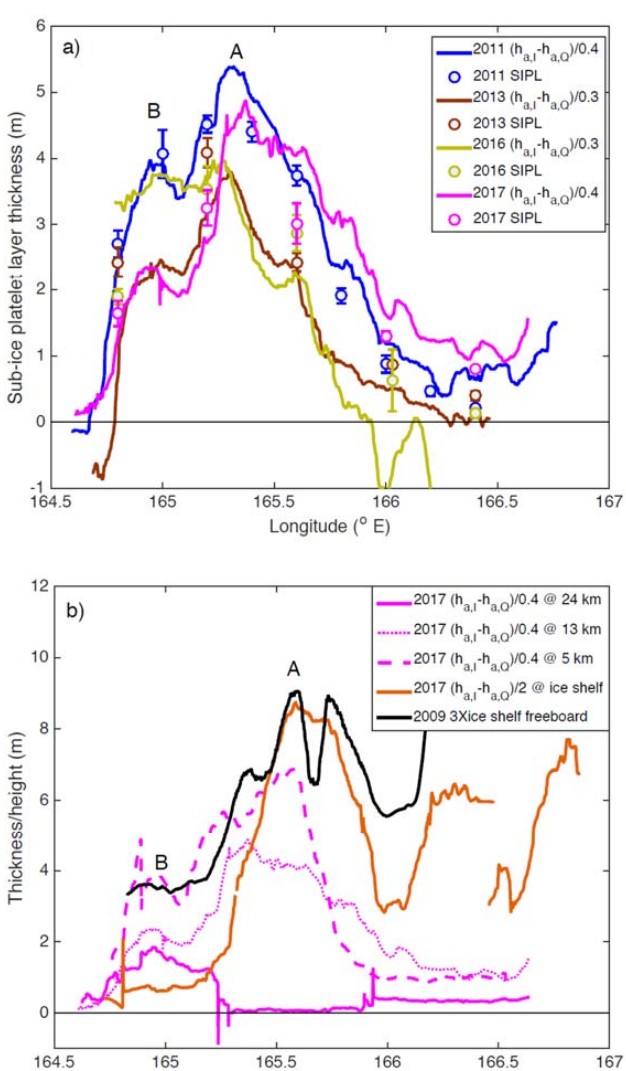

**Figure 10: (a)** SIPL thickness along 77°46'S in 2011, 2013, 2016 and 2017, approximately 11 km from McMurdo Ice Shelf front. Recurring local maxima or shoulders in thickness are identified at A and B. **(b)** SIPL thickness profiles in November 2017 at different distances from McMurdo Ice Shelf front (pink lines), at approximately 24 km (solid), 13 km (dotted) and 5 km (dashed). Figure also shows uncalibrated, scaled SIPL thickness beneath ice shelf in 2017 (brown), and scaled ice shelf freeboard in 2009 (black; from Rack et al 2013), at 77°55' S. The ice shelf data are smoothed by a moving average filter of window size 50. Local thickness maxima are identified at A and B.





For comparison, Figure 10b also includes data from AEM and laser altimeter surveys of the McMurdo Ice Shelf near its front at 77°55' S (see ice shelf locations in Figure 11) in November 2009 and 2017. It shows the ice freeboard in 2009 from Rack et al. (2013) and an uncalibrated measure of the SIPL thickness beneath the ice shelf. The latter was derived from the difference between inphase and quadrature apparent thicknesses, $h_{a,I}$ - $h_{a,Q}$, but no scaling was applied. Note that the ratio between $h_{a,Q}$ and $h_i$, and scaling factor $\alpha$ (Eqs. 5) under the 20 to more than 50 m thick ice shelf could be quite different than under 2 m

thick sea ice, and that no calibration measurements were available.

Figure 10b shows that the locations of the local maxima in SIPL thickness under the ice shelf in 2017 coincide very well with Rack et al.'s (2013) ice shelf freeboard in 2009. This could be due to preferential accretion of marine ice in those locations, or due to the increased buoyancy from the SIPL under the ice shelf (Rack et al., 2013). Even more importantly, the locations of the SIPL thickness maxima under the ice shelf coincide approximately with the locations of SIPL thickness maxima under the

fast ice to the north, providing evidence that the structure of the SIPL under the fast ice is directly linked to the geometry of the ISW outflow from under the ice shelf.

The local maxima A and B illustrated in Figure 10 can be visually identified in some transects from all years 2009, 2011, 2013, 2016 and 2017 and their positions are shown in regional context on the map in Figure 11. The peaks clearly originate under the ice shelf and propagate beneath the sea ice, carried northward by the ISW plume. The thickest peak A appears to be carried

westward, as is also visible in Figure 10. The westward displacement of this peak may be supported by the Coriolis force acting on the northward flowing ISW (Robinson et al, 2014), as suggested by the modeling of Cheng et al (2019) and Holland and Feltham (2005). Peak B is farther west and appears to originate from under the ice shelf near the Koettlitz glacier. Its course is more northerly as it may be constrained by the 200 m isobath. While the ice shelf thickness measurements near the front are uncalibrated, they are generally in agreement with thicknesses from 1960-1984 (McCrae, 1984) and from 2015

(Campbell et al, 2017).

Finally, Figure 12a shows the thickness of peaks A and B from Figure 11 versus latitude. Thicknesses were averaged over a width 0.1° of longitude centered on the peak location to be statistically more representative. Although quite noisy, the figure shows that peak A is generally larger than peak B, and that both are decreasing with distance from the ice shelf front. Peak A decreases from a maximum SIPL thicknesses of 8 m approximately 3 km from the front to less than 3 m at 24 km, i.e. over a

distance of 21 km. The relatively large scatter of up to 2 m at single locations is due to the described interannual variability and retrieval uncertainty. At the northernmost transect 24 km from the ice shelf (77°40'S) only one peak was identifiable. It is quite possible that the converging paths of peaks A and B have merged at that latitude.

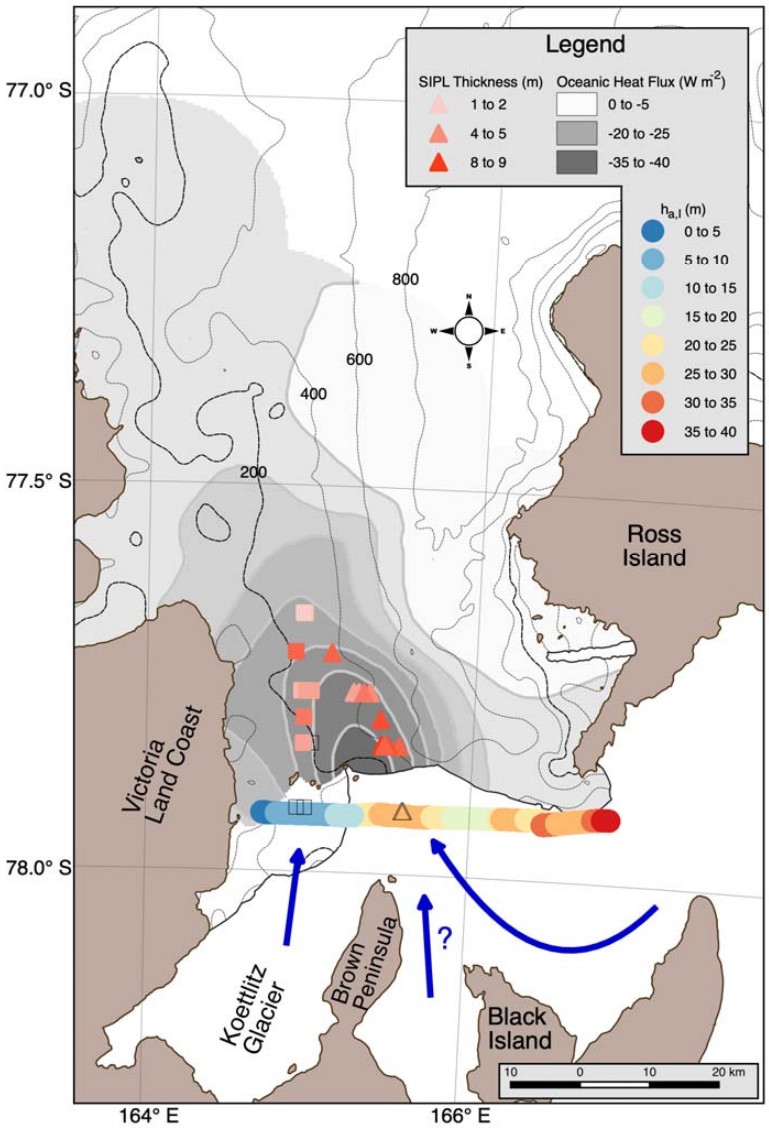

**Figure 11: (a) Bathymetric map of McMurdo Sound showing the location of local maxima in SIPL thickness, with peak A as triangles, and B as squares for all years 2009, 2011, 2013, 2016 and 2017. The magnitude of the local maxima are colored proportionally to the thickness of the peak, with darker orange for thicker average SIPL. Open symbols denote peaks whose absolute thicknesses were uncertain. Colored horizontal line shows ice shelf $h_{a,I}$ profile flown in 2017 with locations of corresponding A and B peaks identified. Contours in grey are proportional to negative ocean heat flux from Langhorne et al. (2015). Blue arrows indicate possible paths of**
**surface ISW (based on Robinson et al, 2014).**





In contrast, Figure 12b shows integrated SIPL thicknesses across the complete individual east-west transects. The integral was calculated for cross-sections with SIPL thicknesses of at least 1 m. A few thickness surveys had to be ended before SIPL thickness decreased below 1 m in the west, near the coast. In these cases data were simply extrapolated following the generally steeply decreasing thickness gradients found in the west (e.g. Fig. 10a). These integral thicknesses are less influenced by the

peak thicknesses, but more representative of the overall volume of SIPL at the different distances from the ice shelf. However, the same general behavior as with peak thicknesses in Figure 12a can be seen, with all peaks decreasing in thickness with distance from the ice shelf, from south to north. The figure also confirms that SIPL thicknesses and therefore volumes were larger in 2011 and 2017 than in 2013 and 2016. These results are in general agreement with SIPL volumes derived from ground-based EM surveys by Brett et al. (2020) also shown in Figure 12b. Note that absolute values are difficult to compare

because Brett et al. (2020) derived their average results from data that were spatially gridded across the central McMurdo Sound.

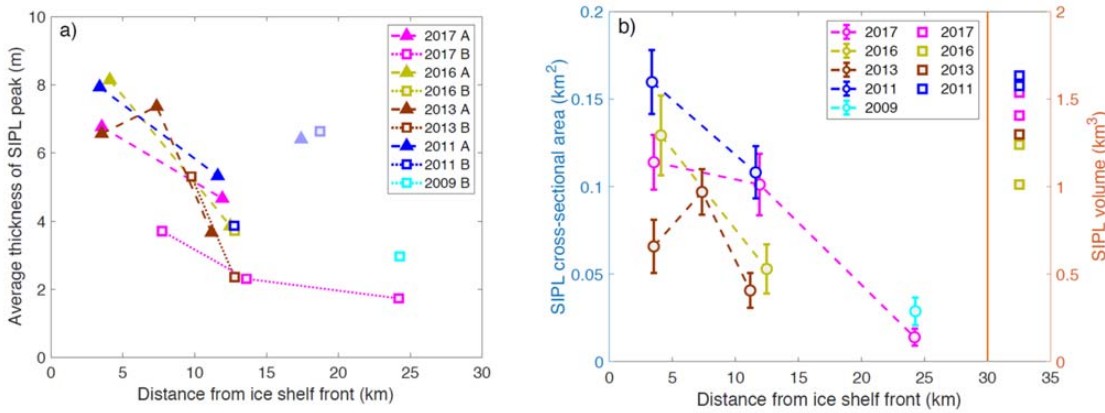

**Figure 12: (a) SIPL thickness of peaks A and B (averaged over 0.1° of longitude) versus distance from ice shelf front for all years**
**2009, 2011, 2013, 2016 and 2017 (cf. Fig. 11). (b) East-west cross-sectional area through the SIPL region (defined as greater than 1 m thickness). Simultaneous SIPL volumes over a 675 km² area in the central McMurdo Sound from Brett et al. (2020) are shown on right (squares). Error bars assume a ± 0.5 m data reduction model uncertainty in EM SIPL measurements (Section 3.2).**

## 4 Discussion

In this study we have presented a new, simple method to map the distribution and thickness of a sub-ice platelet layer (SIPL) by means of airborne EM surveying. The accuracy of the method was assessed with theoretical considerations and by means of comparisons with drill-hole data. Regression of EM results with drill-hole data showed very low bias with a slope of 1 and



intercept of 0 m, but a root mean square error of 0.47 m. This uncertainty is partially due to the EM measurement noise in the order of 10 ppm in $I$ and $Q$, whose effect on retrieved ice thicknesses increases with increasing thickness and decreasing $I$ and
$Q$ signals, due to the negative exponential EM response to increasing ice thickness.

In a few instances, we also observed that the retrieved SIPL thicknesses were actually negative, but still within the derived rms errors (e.g. Fig. 10a). Negative values arise when the inphase derived apparent total thickness is smaller than the quadrature derived apparent consolidated ice thickness (Eq. 5c) which can happen when the SIPL is very thin or absent. The quadrature signals are not only much weaker than the inphase signals (Fig. 3), but are also subject to stronger electronic instrument drift.
This makes the detection of SIPL layers thinner than 0.5 m very challenging.

In addition to uncertainties due to instrument effects, variable SIPL conductivities contribute to variations in the EM response even with constant SIPL thicknesses. In fact our inversions suggest quite a wide range of SIPL conductivities between 900 and 1800 mS/m, which is larger than the relatively narrow estimates of, e.g., 900 to 1400 mS/m by Hunkeler et al. (2015 a&b). This led us to distinguish between different SIPL conductivities of 900 to 1500 mS/m in 2009, 2011, and 2017, and 1000 to
1800 mS/m in 2013 and 2016. These interannual differences also led to different SIPL scaling factors $\alpha$ in the different years, $\alpha = 0.4$ in 2009, 2011, and 2017, and $\alpha = 0.3$ in 2013 and 2016, in agreement with reduced EM sensitivity for higher SIPL conductivities. The separation between these years was only possible due to the availability of drill-hole calibration data. In the absence of drill-hole data the uncertainties would be larger than in this study, and can best be inferred from Figure 5, that showed the range of possible variability of the consolidated ice thickness retrieval $h_{a,Q}/h_i$ and of the SIPL scaling factor $\alpha$.
Consequently, there is a data reduction model uncertainty of 0.5 m that accounts for these simplifying assumptions in the use and choice of $\alpha$.

In 2013 and 2016 we not only found higher SIPL conductivities than in 2009, 2011, and 2017, but those were also the two years with a thinner SIPL (see Figure 12 b). It is intriguing to consider if there is a relationship between thinner SIPL and larger SIPL conductivity, i.e. negative correlation between SIPL thickness and conductivity. Similar behavior was observed
by Hunkeler et al. (2015b) and Hoppmann et al. (2015). They observed negatively correlated SIPL thickness and conductivity with variable SIPL thickness along their profiles surveyed within a few days, while our observations represent spatially averaged, annual conditions obtained over a period of several years. However, the general behavior could potentially be explained by the age of the SIPL or the intensity of SIPL formation in a certain location or year, where more rapid or more massive SIPL formation is caused by more intensive inflow of supercooled ISW under the fast ice, or by longer accumulation
times. Both processes may support more rapid or extensive consolidation of the SIPL interstitial pore space, which increases solid fraction and decreases conductivity, thus causing the observed behavior.

Despite the uncertainties discussed above, our results are in close agreement with the results of Brett et al. (2020), who used ground-based EM surveys to find that SIPL thicknesses in McMurdo Sound were less in 2013 and 2016 than in 2011 and 2017. As Brett et al. (2020) demonstrate, thicker SIPLs occur in years with the occurrence of more frequent strong southerly winds
and hence higher polynya activity.



We provide direct evidence that the ISW plume of McMurdo Sound flows out from beneath the McMurdo Ice Shelf. Our results show consistently that the SIPL extent in the west displays relatively little interannual variability, while variability near its eastern margin is quite large (Figs. 9 and 10). In addition, east-west SIPL thickness gradients are greater in the west than in the east. As the SIPL structure and thickness are closely related to the properties of the outflowing ISW to the south, we agree with Robinson et al (2014) that the ISW outflow from the McMurdo Ice Shelf in the west is strongly controlled by bathymetry and the fact that the western margin is close to the coast and constrained by shallow water (Jendersie et al., 2018). The location of the western peak of SIPL thickness at water depths of around 200 m (Fig. 11) suggests that the currents driving the ISW plume are constrained by bathymetry there (Robinson et al, 2014). In contrast, in the east the ISW structure is dependent on the interplay with the warmer and more saline water inflowing from the north on the eastern side of McMurdo Sound (Leonard et al, 2006; Mahoney et al, 2011; Leonard et al, 2011; Robinson et al, 2014). The interplay controls both the extent and thickness of the SIPL in eastern McMurdo Sound. The source of the ISW outflow is the Ross/McMurdo Ice Shelf (Robinson et al, 2014; Jendersie et al, 2018) and there are a number of possible explanations for the two local peaks observed in the SIPL thickness. The first is that the two streams arise from different sources: Robinson et al, (2014) suggested that one local maxima (peak B) may be from the Koettlitz Glacier which has retreated for over 100 years (Gow and Govoni, 1994). The larger maximum, peak A, likely originates from the confluence of the McMurdo and Ross Ice Shelves (Robinson et al, 2014) as indicated by the arrow north of Black Island in Figure 11. Alternatively, marine ice has been found in the southern McMurdo Ice Shelf (Koch et al, 2015; Grima et al, 2019) and a possible additional source may emerge from the channel between Black Island and the Brown Peninsula (see Figure 1a and 11). Once it emerges from under the ice shelf and spreads out under the fast ice, this stream moves westward under the influence of the Coriolis force (Robinson et al, 2014) as modelled by Cheng et al. (2019). Alternatively, it may be that there is one ISW outflow that is split by sea floor and ice shelf morphology and islands close to ice shelf front. More concurrent oceanographic and EM surveys are required to further study this interplay within the coastal current that flows northward up the coast of Victoria Land.

## 5 Conclusions

We have presented results from five AEM ice thickness surveys of the landfast ice in McMurdo Sound in November of 2009, 2011, 2013, 2016, and 2017 with the aim to describe the spatial and interannual variability of the sub-ice platelet layer (SIPL) known to exist below the fast ice. We have presented a simple method to obtain approximate SIPL thickness and conductivity information from the inphase and quadrature components of single-frequency AEM data, that was calibrated and validated with drill-hole measurements. Forward EM modeling demonstrated the varying sensitivity and accuracy of the method over ice with variable thickness and underlain with a SIPL with variable thickness and conductivity. Results are in good agreement with previous knowledge of the SIPL distribution, thickness, and conductivity and solid fraction in McMurdo Sound. However, the extensive, continuous data with high spatial resolution that are possible with airborne surveys, provided new insights into the small-scale spatial variability of SIPL thickness, and in particular provide novel evidence for the presence of at least two



elongated regions of thicker SIPL that may bear information about the structure of the Ice Shelf Water (ISW) plume. We were able to show that the spatial occurrence of those thicker SIPL regions closely corresponds to thickness and SIPL occurrence

under the ice shelf, thus linking processes under the ice shelf with the structure of the SIPL under the landfast ice.

The association of the SIPL with ISW and its link to melting and circulation processes under ice shelves makes our approach particularly attractive for exploratory mapping of the vast, remote regions of fast ice fringing the circum-Antarctic ice shelves. We could easily discover the occurrence and thickness of a SIPL in these unstudied regions. Variations in the thickness of the SIPL are indicators of intensive, near-surface ISW outflow in response to ice shelf bottom melt. Such mapping could therefore

identify potential "hotspots" of present basal ice shelf melt, and could provide important advance information for subsequent future, more comprehensive ice-shelf/ocean studies. The network of circumpolar coastal Antarctic research stations and their airfields makes it entirely feasible to carry out such survey with Basler aircraft that are used by many national Antarctic research programs.

### Acknowledgement

We are most grateful for the logistics support and aircraft funds provided by Antarctica NZ and the welcoming staff at Scott Base. We particularly thank Johno Leitch and his team for excellent ground support for the Basler BT67 airplane campaigns in 2016 and 2017. The success of this project would not have been possible without the dedication of Helicopter NZ pilot Rob McPhail, Southern Lakes Helicopters pilot Hannibal Hayes, the Kenn Borek Air BT67 captains Gary Murtsell and Jamie Chisholm, and their respective air and ground crews. Field logistics and air time were funded by *Targeted observations and*

*process-informed modeling of Antarctic sea ice* through the Deep South National Science Challenge + K053 K063 (2009, 2013, 2011). CH acknowledges infrastructure and operation funding by Alberta Ingenuity Scholarship grant AITFschoptg_200700043_Haas, Tier 1 Canada Research Chair grant # 950-228139, and NSERC Discovery Grant #356589.

### Data availability

All data will be made available at the World Data Center PANGAEA https://www.pangaea.de/

### Author contribution

CH, PJL, and WR designed the field experiments, analyzed the data, developed the retrieval algorithm, and secured the required funds. All authors contributed to field data acquisition and contributed to writing of the manuscript.

### Competing interests

All authors declare no competing interests.



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
