# Peer review of "Airborne mapping of the sub-ice platelet layer under fast ice in McMurdo Sound, Antarctica"

_The Cryosphere, 2020_

## Referee Comment (RC1) · Blake P. Weissling (Referee) · 25 Oct 2020

This paper describes a novel yet simple approach to derive the presence, thickness, and spatial distribution of the sub-ice platelet layer (SIPL) under McMurdo Sound consolidated sea ice through forward modeling of airborne electromagnetic induction (inphase and quadrature component) survey data regressed with drill-hole measurements. The underlying premise of the approach is based on the well-documented (since the 1960's) approach of electromagnetic induction (EM) sounding measurements of ice thickness to ice and sea water conductivity under half-space considerations. However, the presence of a platelet layer under consolidated sea ice challenges the simple two-layer (near non-conductive sea ice over highly conductive sea water) model considerations. This new model is based upon an apparent SIPL thickness

derived from a much more robust forward model application of a Hankel transform of sea ice EM reflection coefficients, electrical conductivity, magnetic permeability, and layer thickness, as well as survey/instrument geometry such as coil spacing, instrument height, and operating frequency. Model derived apparent thickness of the SIPL is then calibrated against drillhole-derived thickness, yielding a scaling factor to derive true thickness. The scaling factor appears to be somewhat variable depending on intra-annual ice conditions driving variability in assumptions of bulk conductivity of the SIPL ice crystal-sea water mixture.

General Comments: This referee's opinion of the author's approach, mathematical model, results, and presentation is quite positive. Having some practical knowledge myself in EM sounding of ice thickness, with awareness of the latest research in this geophysical endeavor, I rate the originality of the approach as excellent. The manuscript, likewise, is very readable and clear in its development of the premises of the model and the application toward this very important development in mapping the presence, thickness, and physical properties of the SIPL. The methods are rigorous but straightforward and understandable, meeting the needs of both researchers and lay-men interested in the topic. This manuscript gets my full endorsement for publication in TC with only some minor revision as described below in my specific comments.

Specific Comments: In section 2.1.2 (line 167), the authors state that the conductivity of the consolidated ice in this layered model is set at 0 ms/m (infinite resistivity). Such a conductivity would assume that all entrained brine has been drained from the sea ice and I'm wondering if such an assumption is valid. Even with very cold first year ice, I would assume that some brine pockets remain albeit not necessarily connected in a significantly porous permeable network. Could the author's comment on this and if you considered a non-zero conductivity in your model (say at < 50 ms/m bulk ice conductivity)? In section 2.1.3, the author's present a mathematical model based on a continuous integration (Hankel transform). Is there a discrete version of this transform, and if so how was it computationally implemented? I'm not looking for an elaborate

explanation, but just a short description of what the discrete transform looks like and if it was solved/applied in Matlab or some other software package. This would be useful for other researchers in applying the method. Perhaps an addenda describing the computational approach in more detail. Saves us from having to reinvent the wheel so to speak. In section 3.3, paragraph 2, the authors describe only the cementation factor in the application of Archie's Law. There are 2 other factors or coefficients relevant to Archie's Law ( )for brine saturated porous media, the tortuosity factor (a) and the saturation exponent (n). Could the author's address this and if such coefficients/exponents have been formulated for ice media? Both numbers I suspect would be relevant in an ice matrix.

Technical comments: In section 2.1 (line 108-9), the sentence beginning "The surveys covered . . ." could be rephrased as it reads a bit awkward. In line 117, is "levelness" best word choice? In that same line, could the author's elaborate briefly on "occasional noise" and what it entails? In section 2.1.3 (line 254) do you mean to say "derived apparent conductivities" or "derived apparent thicknesses"? This possible typo is repeated in line 271, "from the apparent (conductivity) of the Q measurement". Please check as I think you meant to say "thickness" in both cases. In section 2.2 (line 316), "we assume of typical" is strange wording. Maybe drop the "of".

---

## Referee Comment (RC2) · Andrew Mahoney (Referee) · 28 Oct 2020

**Summary**

This manuscript presents a comprehensive analysis of airborne electromagnetic (AEM) survey data over sea ice in the McMurdo Sound including a semi-empirical approach for measuring the both the thickness of the consolidated sea ice and the depth of the sub-ice platelet layer (SIPL) below it. Typically, the presence of a SIPL complicates the AEM-based measurement of ice thickness since it effectively introduces another independent variable that affects the observed apparent conductivity upon which the ice thickness measurement is based. However, using a forward modelling approach, the authors demonstrate that the quadrature (Q) component of the induced AEM signal

is less sensitive to the presence of a SIPL than the in-phase (I) component.

On this basis, the authors determine that the SIPL thickness can be estimated from the difference in the apparent thicknesses determined from the I- and Q-components. And using drill-hole observations of the thickness of both consolidated ice and the SIPL measured at discrete points beneath AEM surveys, they show that the thickness of the SIPL in McMurdo is approximately 0.4 times the difference between the I- and Q-derived apparent thickness measurements. This factor of 0.4 is related to the effective conductivity of the SIPL, which is determined in part by the solid fraction of ice within the SIPL. Using this calibrated relationship, the manuscript presents measurements of SIPL thickness from 5 years of AEM surveys in McMurdo Sound. The data show a persistent, interannual pattern of enhanced SIPL thickness in two locations, indicating the likely outflow path of ice shelf water (ISW) from beneath the McMurdo Ice Shelf. The production and pathways of ISW are closely tied to basal melt processes and the authors conclude by noting the value of AEM surveying for mapping and monitoring SIPL thickness and ISW flow in other remote regions of the Antarctic coast.

This is a well written paper and easy to read paper summarizing several years of work that has culiminated in a greater understanding of platelet distribution in McMurdo Sound that has implications for ISW formation and basal melt. At the same time, the technique used could applied with great value in other locations. I have only one general comment, that I think should be fairly straightforward to address. Otherwise, I look forward to seeing this paper published.

**General Comments**

Limited discussion on variability in consolidated in thickness

My only major comment relates to the observation that the data presented represent a very narrow range of consolidated ice thicknesses, $h_i$. Figure 5a indicates the ratio $h_{(a,Q)}/h_i$ increases as $h_i$ decreases, which leaves me wondering how reliable the SIPL thickness estimates would be in the presence of greater spatial or temporal variability

in $h_i$. It is unfortunate that only few measurements were made in thicker multiyear ice, but the inclusion of a brief discussion about how they compare with the AEM measurements might still be helpful. The authors might also consider how the sensitivity to SIPL thickness would be affected over thin ice, earlier in the year, which could be relevant for studying intra-annual SIPL variability

**Minor Comments**

Line 175-176: I recommend introducing the concept of apparent thickness at the beginning of this paragraph and the symbol $h_a$ should be defined before it first used.

Line 176: I don't think the parentheses around "or Q" are necessary

Line 400: If the approximate values of the dimensions a,b,c, and d are really important to this discussion, then I feel the reader should be given some further explanation without need to refer to the paper by Jones 2012b. At the least, the text should provide units for the stated values.

---

## Editor Comment (EC1) · Ted Maksym (Editor) · 30 Oct 2020

In addition to the minor points raised by the two reviewers, I would like to add to the comment from RC2 about the accuracy in derived SIPL for different hi. I note that for the thicker SIPL layers ($\sim$4-8 m), assuming a conductivity in the middle of the estimated range ($\sim$1200 mS/m), that the behaviour upon which eq. 5a-c are based does not really hold. The decrease in I with increasing SIPL levels out, and Q shows a slight, increase, comparable to the variability in I (Fig 3). Hence, $h_{a,I}$ has a modest increase as SIPL increases between 4-8m that is comparable to the decrease in $h_{a,Q}$ over the same range. But Fig 6 is very convincing that eq 5b and 5c work. I think this is because the relationship between $h_{a,Q}$ and SIPL is roughly linear over this range, so that 5c remains linear, but would slightly modify the effective value of alpha. It might be that since the

effect is modest, this relationship still works well enough over the range of SIPL. I do note that I'd expect this to reduce alpha for larger SIPL (if I have the direction of the effect correct), which is not consistent with the lower value used for 2013 and 2016, so it seems it is not a large enough effect to make much difference to the empirical fits. However, similar to RC2's question, I wonder if this effect may become important if the consolidated ice thickness varied significantly?

As noted, the effect seems to be unimportant, so I do not necessarily suggest any major changes need be made; ultimately eq. 5c is semi-empirical, and the effect of variability in ha,Q is not so important. However, for these ranges of SIPL and conductivity, neither I or Q vary much (figure 3). This would suggest that the derived SIPL values may have quite large uncertainty. Can you provide some estimate of uncertainty due to the precision and/or accuracy of the instrument for these SIPL thicknesses? Given figure 4a, it seems that for SIPL thicknesses above ∼5-6m, that the SIPL estimate would be essentially indeterminate. Particularly since relatively slight variations in conductivity seem to have as great, or greater effect. Given this, it is perhaps somewhat surprising that the SIPL thicknesses match observations as well as they do (although many of the estimated SIPL thicknesses are well outside the drill hole error bars in Figs 9 and 10). Some further discussion of potential limitations of this method based on this may be helpful to add to the text.

A couple other suggestions (up to authors' discretion):

Figure 1: Might it be better for the green dots to show the total estimated thickness (consolidated + SIPL), or just the SIPL thickness, rather than the apparent thickness (which is not an actual thickness)? A casual reader might look carefully only at this figure, as it provides a clear geographical presentation of the results. I think it would be best if this showed the actual results.

Figure 6: it might be useful to show the full drill hole thicknesses here as well (consolidate + SIPL) so that the scaling factor is more apparent.

[Figure]

---

## Author Comment (AC1) · 22 Nov 2020

Replies to Reviewer 1 (Blake P. Weissling) comments

We are very grateful for the very positive evaluation by this reviewer, and by the suggestions for further improvements, which we have considered carefully. Please find our replies (AC) to the specific reviewer comments (RC) below:

RC: In section 2.1.2 (line 167), the authors state that the conductivity of the consolidated ice in this layered model is set at 0 ms/m (infinite resistivity). Such a conductivity would assume that all entrained brine has been drained from the sea ice and I'm wondering if such an assumption is valid. Even with very cold first year ice, I would assume that some brine pockets remain albeit not necessarily connected in a significantly porous permeable network. Could the author's comment on this and if you considered a non-zero conductivity in your model (say at < 50 ms/m bulk ice conductivity)?

AC: The reviewer is right in stating that the conductivity of consolidated sea ice can be larger than zero due to connected brine residing within the ice. However, conductivities between 0 and 50 mS/m do not affect the EM thickness retrieval much (less than 10 cm). We have therefore added the likely range of consolidate ice conductivity to Section 2.1.2 "(~0-50 mS/m; Haas et al., 1997)" and have added a sentence to the end of the model description of Section 2.1.3:

"Note that we chose $\sigma_i$ = 0 mS/m for simplicity, while in reality consolidated sea ice still contains some brine that can slightly raise its conductivity up to $\sigma_i$ = 50 mS/m or so (Haas et al., 1997). However, those small variations have little effect on the EM retrieval of consolidated ice thickness (Haas et al., 1997; 2009)."

RC: In section 2.1.3, the author's present a mathematical model based on a continuous integration (Hankel transform). Is there a discrete version of this transform, and if so how was it computationally implemented? I'm not looking for an elaborate explanation, but just a short description of what the discrete transform looks like and if it was solved/applied in Matlab or some other software package. This would be useful for other researchers in applying the method. Perhaps an addenda describing the computational approach in more detail. Saves us from having to reinvent the wheel so to speak.

AC: We thought that the stated references provided enough information on the modeling, but agree that stating a few more details in our manuscript would clarify the modelling more immediately. Indeed the model cannot be computed analytically but requires a numerical method that uses digital filters. While we were using old software which I developed during my PhD and that was based on Anderson (1979), one of my former PhD students has indeed published a matlab based code on Pangaea:

Irvin, A.: One Dimensional Frequency domain Electromagnetic Model (ODFEM). PANGAEA, https://doi.org/10.1594/PANGAEA.897352, 2019.

Anderson, W. L: Computer Program. Numerical integration of related Hankel transforms of orders 0 and 1 by adaptive digital filtering, Geophysics, 44, 1287–1305, https://doi.org/10.1190/1.1441007, 1979.

In the manuscript we have added a sentence immediately below the first mention of the Hankel Transform:

"This equation can only be solved numerically using digital filters. Here we used the filter coefficients of Guptasarma and Singh (1997) that are, for example, implemented in a program by Irvin (2019)."

RC: In section 3.3, paragraph 2, the authors describe only the cementation factor in the application of Archie's Law. There are 2 other factors or coefficients relevant to Archie's Law ( )for brine saturated porous media, the tortuosity factor (a) and the saturation exponent (n). Could the author's address this and if such coefficients/exponents have been formulated for ice media? Both numbers I suspect would be relevant in an ice matrix.

AC: These are good points. Honestly we are not sure what the latest studies about sea ice conductivity are, except for the work of Jones which we use later in our manuscript and that is less based on using Archie's law. However, our approach largely follows the work by Haas and Hunkeler which we clearly cite. In that work tortuosity and saturation are assumed to be one, for a lack of better information. These assumptions also follow yet earlier work by Kovacs and Morey (1986). We have now added that reference and added the following comment:

Figure 8 shows the horizontal conductivity for Archie's Law "with tortuosity factor and saturation exponent set to one (e.g. Kovacs and Morey, 1986)", and cementation factor m=1.75 (Haas et al., 1997), and m=3 (Hunkeler et al., 2015b),

Kovacs, A., and Morey, R.M.: Electromagnetic measurements of multi-year sea ice using impulse radar, Cold Regions Science and Technology, 12(1), 67-93, https://doi.org/10.1016/0165-232X(86)90021-2, 1986.

RC: Technical comments: In section 2.1 (line 108-9), the sentence beginning "The surveys covered : : :" could be rephrased as it reads a bit awkward.

AC: OK

RC: In line 117, is "levelness" best word choice? AC: We reconsidered the word but find it quite suitable in this context.

RC: In that same line, could the author's elaborate briefly on "occasional

noise" and what it entails?

AC: added: noise from EMI interference or episodic electronic drift

RC: In section 2.1.3 (line 254) do you mean to say "derived apparent conductivities" or "derived apparent thicknesses"? This possible typo is repeated in line 271, "from the apparent (conductivity) of the Q measurement". Please check as I think you meant to say "thickness" in both cases.

AC: Thank you for your careful reading, yes these should be thicknesses.

RC: In section 2.2 (line 316), "we assume of typical" is strange wording. Maybe drop the "of".

AC: thanks, yes the "of" has to go...;)

---

## Author Comment (AC2) · 22 Nov 2020

Replies to Reviewer 2 (Andrew Mahoney) comments

AC: We are very grateful for the very positive evaluation by this reviewer, and by the suggestions for further improvements, which we have considered carefully. Please find our replies (AC) to the specific reviewer comments (RC) below:

RC: This is a well written paper and easy to read paper summarizing several years of work that has culminated in a greater understanding of platelet distribution in McMurdo Sound that has implications for ISW formation and basal melt. At the same time, the technique used could applied with great value in other locations. I have only one general comment, that I think should be fairly straightforward to address. Otherwise, I look forward to seeing this paper published.

General Comments

Limited discussion on variability in consolidated in thickness

My only major comment relates to the observation that the data presented represent a very narrow range of consolidated ice thicknesses, hi. Figure 5a indicates the ratio h(a;Q)=hi increases as hi decreases, which leaves me wondering how reliable the SIPL thickness estimates would be in the presence of greater spatial or temporal variability in hi. It is unfortunate that only few measurements were made in thicker multiyear ice, but the inclusion of a brief discussion about how they compare with the AEM measurements might still be helpful. The authors might also consider how the sensitivity to SIPL thickness would be affected over thin ice, earlier in the year, which could be relevant for studying intra-annual SIPL variability.

AC: We very much appreciate this comment and indeed were going to address this much more clearly in the initial manuscript but then it got lost... Therefore we have added two paragraphs to the discussion that address this reviewer's comment as well as those from the editor:

"Our validation data are limited to drill-hole measurements from first-year fast ice that is typically 2 m thick at the end of the winter. Therefore, most of our model results were also limited to 2 m thick consolidated ice. However, Figure 5 also includes results for 4 m and 6 m thick consolidated ice (dashed curves). From the behavior of those model curves it can be inferred that with thicker consolidated ice the ratio of ha,Q/hi decreases, which suggests that, in the presence of a typical SIPL, thicker consolidated ice can be retrieved more accurately than thinner ice from the quadrature measurements. Figure 5 also shows that the scaling factor α is hardly affected by consolidated ice thickness at all, i.e. the accuracy of retrieved SIPL thicknesses is independent of ice thickness. The thickness profiles in Figure 9a include surveys of multiyear fast ice in 2013 and 2017, which are visible by large steps towards thicker ice in the west. These are indications that the measurements are indeed quite sensitive to thicker consolidated ice and SIPL as well. We only attempted very few drill-hole measurements of the thick consolidated ice and thick SIPL, as they are very challenging and their accuracy is poor. Therefore we did not include them in our analysis here.

However, thick consolidated ice and a thick SIPL pose other challenges that are related to the decreasing sensitivity of EM measurements with increasing height above the water or conductive SIPL. Despite the

better behavior of ha,Q/hi discussed above with regard to Figure 5, thicker consolidated ice results in weaker inphase and quadrature signals which eventually approach the EM noise level and are then insensitive to consolidated ice thickness changes (not shown here, see Haas et al., 2009). However, these limitations only apply to ice several tens or meters thick (e.g. Rack et al., 2013). More importantly, increasing SIPL thicknesses also lead to reduced sensitivities particularly of the inphase signals as has been discussed above with regard to results shown in Figure 3. That figure shows that for typical SIPL conductivities of 900 mS/m and more the inphase signal remains approximately constant for SIPL thickness of 6 m and more. This is due to the limited EM field depth penetration into conductive layers, which make the method insensitive to changes below the level of penetration. Therefore it is likely that the good results shown in Figure 7 benefited from the fact that most drill-hole SIPL thicknesses in the study region were not larger than 6 m (total thickness of 8 m). In fact, Figure 7 shows that the uncertainties of the thickest SIPL measurements which also have the largest drill-hole errors are considerably larger than those of smaller total thicknesses."

RC: Line 175-176: I recommend introducing the concept of apparent thickness at the beginning of this paragraph and the symbol ha should be defined before it first used.

AC: agreed, we have re-arranged the sentences in that paragraph.

RC: Line 176: I don't think the parentheses around "or Q" are necessary

AC: removed!

RC: Line 400: If the approximate values of the dimensions a,b,c, and d are really important to this discussion, then I feel the reader should be given some further explanation without need to refer to the paper by Jones 2012b. At the least, the text should provide units for the stated values.

AC: As the text stated, a, b, c are relative brine pocket dimensions and therefore have no units. However we agree that some further explanations may be useful and have therefore slightly extended the whole paragraph:

"For sea ice specifically, Jones et al. (2012a) have used a simple conductivity model (Jones et al., 2012b) to derive parameters for an ice/brine "unit cell". Each unit cell consists of a single, isolated, cubical brine pocket with sides of relative dimension d (unitless), and three connected channels in perpendicular directions (two horizontal and one vertical direction), each with relative dimensions c    a    b (also unitless). Jones et al (2012b) found that the relative dimensions that fit the observed in situ DC horizontal and vertical resistivities depend not only on sea ice temperature but also on structure. In particular, for Antarctic incorporated platelet ice at   5°C the shape of the inclusions has relative brine pocket dimensions a≈1, b≈17, c≈0.6, and d≈6 (see Jones et al. (2012a) for details). In addition, Jones et al (2012b) have shown for Arctic first-year sea ice that there is a dramatic change in these parameters with temperature, with a, c, and d becoming relatively larger, while b drops. This behavior would also be expected in incorporated platelet ice. We shall therefore assume that the shape of the inclusions in the SIPL is similar to that of incorporated platelet ice (as observed by Jones et al., 2012a), but that brine

inclusion/void dimensions are very much larger because they are very close to the freezing point. Consequently, we calculate the relationship between solid fraction and conductivity from Jones et al. (2012b), by varying a and c, while keeping b and d constant and hence changing the solid/liquid content of the SIPL (see Fig. 8)."

---

## Author Comment (AC3) · 22 Nov 2020

Replies to Editor (Ted Maksym) comments

AC: Thank you for your additional comments Ted, which help to clarify our manuscript further. Please find our replies (AC) to the specific editor comments (EC) below:

EC: In addition to the minor points raised by the two reviewers, I would like to add to the comment from RC2 about the accuracy in derived SIPL for different hi. I note that for the thicker SIPL layers ( 4-8 m), assuming a conductivity in the middle of the estimated range ( 1200 mS/m), that the behaviour upon which eq. 5a-c are based does not really hold. The decrease in I with increasing SIPL levels out, and Q shows a slight, increase, comparable to the variability in I (Fig 3). Hence, ha,I has a modest increase as SIPL increases between 4-8m that is comparable to the decrease in ha,Q over the same range. But Fig 6 is very convincing that eq 5b and 5c work. I think this is because the relationship between ha,Q and SIPL is roughly linear over this range, so that 5c remains linear, but would slightly modify the effective value of alpha. It might be that since the effect is modest, this relationship still works well enough over the range of SIPL. I do note that I'd expect this to reduce alpha for larger SIPL (if I have the direction of the effect correct), which is not consistent with the lower value used for 2013 and 2016, so it seems it is not a large enough effect to make much difference to the empirical fits. However, similar to RC2's question, I wonder if this effect may become important if the consolidated ice thickness varied significantly?

As noted, the effect seems to be unimportant, so I do not necessarily suggest any major changes need be made; ultimately eq. 5c is semi-empirical, and the effect of variability in ha,Q is not so important. However, for these ranges of SIPL and conductivity, neither I or Q vary much (figure 3). This would suggest that the derived SIPL values may have quite large uncertainty. Can you provide some estimate of uncertainty due to the precision and/or accuracy of the instrument for these SIPL thicknesses? Given figure 4a, it seems that for SIPL thicknesses above  5-6m, that the SIPL estimate would be essentially indeterminate. Particularly since relatively slight variations in conductivity seem to have as great, or greater effect. Given this, it is perhaps somewhat surprising that the SIPL thicknesses match observations as well as they do (although many of the estimated SIPL thicknesses are well outside the drill hole error bars in Figs 9 and 10). Some further discussion of potential limitations of this method based on this may be helpful to add to the text.

AC: Thank you very much for your comments. Indeed, as already expressed in our reply to reviewer 2, these issue were high on our mind initially but were forgotten in the final stages of writing the discussion… Therefore we have added two paragraphs to the discussion that address your and that reviewer's comments as they are closely related. They concern both thick and thin consolidated ice as well as an increasingly thick SIPL thicker than approximately 6 m:

"Our validation data are limited to drill-hole measurements from first-year fast ice that is typically 2 m thick at the end of the winter. Therefore, most of our model results were also limited to 2 m thick consolidated ice. However, Figure 5 also includes results for 4 m and 6 m thick consolidated ice (dashed curves). From the behavior of those model curves it can be inferred that with thicker consolidated ice the ratio of ha,Q/hi decreases, which suggests that, in the presence of a typical SIPL, thicker consolidated ice can be retrieved more accurately than thinner ice from the quadrature measurements. Figure 5 also shows that the scaling factor α is hardly affected by consolidated ice thickness at all, i.e. the accuracy of retrieved SIPL thicknesses is independent of ice thickness. The thickness profiles in Figure 9a

include surveys of multiyear fast ice in 2013 and 2017, which are visible by large steps towards thicker ice in the west. These are indications that the measurements are indeed quite sensitive to thicker consolidated ice and SIPL as well. We only attempted very few drill-hole measurements of the thick consolidated ice and thick SIPL, as they are very challenging and their accuracy is poor. Therefore we did not include them in our analysis here.

However, thick consolidated ice and a thick SIPL pose other challenges that are related to the decreasing sensitivity of EM measurements with increasing height above the water or conductive SIPL. Despite the better behavior of $h_a$,Q/$h_i$ discussed above with regard to Figure 5, thicker consolidated ice results in weaker inphase and quadrature signals which eventually approach the EM noise level and are then insensitive to consolidated ice thickness changes (not shown here, see Haas et al., 2009). However, these limitations only apply to ice several tens or meters thick (e.g. Rack et al., 2013). More importantly, increasing SIPL thicknesses also lead to reduced sensitivities particularly of the inphase signals as has been discussed above with regard to results shown in Figure 3. That figure shows that for typical SIPL conductivities of 900 mS/m and more the inphase signal remains approximately constant for SIPL thickness of 6 m and more. This is due to the limited EM field depth penetration into conductive layers, which make the method insensitive to changes below the level of penetration. Therefore it is likely that the good results shown in Figure 7 benefited from the fact that most drill-hole SIPL thicknesses in the study region were not larger than 6 m (total thickness of 8 m). In fact, Figure 7 shows that the uncertainties of the thickest SIPL measurements which also have the largest drill-hole errors are considerably larger than those of smaller total thicknesses."

In addition, we have added error bars of +/-0.5 m thickness to the graphs in Figures 9 and 10 to illustrate the uncertainty of the AEM retrievals and to show that within that uncertainty the agreement with the drill-hole data is really good.

EC: A couple other suggestions (up to authors' discretion):

Figure 1: Might it be better for the green dots to show the total estimated thickness (consolidated + SIPL), or just the SIPL thickness, rather than the apparent thickness (which is not an actual thickness)? A casual reader might look carefully only at this figure, as it provides a clear geographical presentation of the results. I think it would be best if this showed the actual results.

AC: We have debated various options when we first compiled this figure, including your suggestion. However, we concluded and are still convinced that we should leave it as it is, for the following reasons: 1. In the context of the paper the figure shows the initial apparent thickness that would be obtained when carrying out the measurements. They already show the coherent apparent thickening in the region of the known ISW plume which is the key point of the figure. This was an important discovery/recognition for us when we looked at the data for the first time after our first surveys. 2. Showing the consolidated ice plus SIPL thicknesses would strongly increase the range of displayed values and more subtle details would be less clearly visible. Note also that we only show the surveys over the first-year ice, as the multiyear ice and its SIPL would be much thicker, again increasing the range of displayed values and decreasing their resolution.

However, we agree that it would also be useful to show the full range of measurements, and therefore we have implemented your other suggestion below.

EC: Figure 6: it might be useful to show the full drill hole thicknesses here as well (consolidate

+ SIPL) so that the scaling factor is more apparent.

AC: We agree that this looks interesting as well, and will replace the figures as shown below, even though the scales get much expanded and details of the blue and orange curves are less clearly visible than before.